# A multi-frequency whole-brain neural mass model with homeostatic feedback inhibition

Carlos Coronel-Oliveros[1,2,3☉], Fernando Lehue[4☉], Rubén Herzog[5], Iván Mindlin[6], Marilyn Gatica[7,8], Natalia Kowalczyk-Grębska[9], Vicente Medel[10], Josephine Cruzat[1], Raul Gonzalez-Gomez[1], Hernán Hernandez[1], Enzo Tagliazucchi[1], Pavel Prado[11], Patricio Orio[12,13]*, Agustín Ibáñez[1,2,3,14,15]*

1 Latin American Brain Health Institute (BrainLat), Universidad Adolfo Ibañez, Santiago, Chile, 2 Trinity College Dublin, The University of Dublin, Dublin, Ireland, 3 Global Brain Health Institute (GBHI), Trinity College Dublin, Dublin, Ireland, 4 Advanced Center for Electrical and Electronic Engineering, Universidad Técnica Federico Santa María, Valparaíso, Chile, 5 Department of Psychology, University of the Balearic Islands, Palma de Mallorca, Spain, 6 Institut du Cerveau - Paris Brain Institute - ICM, Sorbonne Université, Inserm, CNRS, Paris, France, 7 Network Science Institute, School of Medicine, Northeastern University London, London, United Kingdom, 8 Precision Imaging, School of Medicine, University of Nottingham, Nottingham, United Kingdom, 9 Faculty of Psychology, SWPS University of Social Sciences and Humanities, Warsaw, Poland, 10 Faculty of Biological Sciences, Pontifical Catholic University of Chile, Santiago, Chile, 11 Escuela de Fonoaudiología, Facultad de Ciencias de la Rehabilitación y Calidad de Vida, Universidad San Sebastián, Santiago, Chile, 12 Centro Interdisciplinario de Neurociencia de Valparaíso (CINV), Universidad de Valparaíso, Valparaíso, Chile, 13 Instituto de Neurociencia, Facultad de Ciencias, Universidad de Valparaíso, Playa Ancha, Chile, 14 Barcelonaβeta Brain Research Center (BBRC), Pasqual Maragall Foundation, Barcelona, Spain, 15 Department of Biophysics, School of Medicine, Istanbul Medipol University,  Istanbul, Türkiye

☉ These authors contributed equally to this work.
* patricio.orio@uv.cl (PO); agustin.ibanez@gbhi.org (AI)

## Abstract

Whole-brain models are valuable tools for understanding brain dynamics in health and disease by enabling the testing of causal mechanisms and identification of therapeutic targets through dynamic simulations. Among these models, biophysically inspired neural mass models have been widely used to simulate electrophysiological recordings, such as MEG and EEG. However, traditional models face limitations, including susceptibility to over-saturation of the sigmoid function by model hyperexcitability, which constrains their ability to capture the full richness of neural dynamics. Here, we thoroughly characterize a previously introduced multi-frequency Jansen-Rit neural mass model with inhibitory synaptic plasticity (ISP) aimed at overcoming these limitations. The ISP adjusts inhibitory feedback onto pyramidal neurons to clamp their firing rates around a target value. This mechanism allows for fine control of neuronal firing rates, preventing over-saturation in whole-brain simulations. In this model, we analyzed how different model parameters modulate oscillatory frequency and connectivity. As a demonstration, we considered simultaneously fitting EEG and fMRI recordings during NREM sleep. Bifurcation analysis showed that ISP widened the range of parameters in which the model exhibited sustained oscillations; the target firing rate can modulate oscillatory dynamics, producing different oscillatory regimes, from slower (δ, θ and α) to faster (β and γ) oscillations. High-frequency activity emerged from low global

**Data availability statement:** All the codes and basic data used to perform the simulations are freely available in GitHub at https://github.com/carlosmig/EEG-Dementias ("JR ReDLat + Sleep" folder), and mirrored in Zenodo (DOI: https://doi.org/10.5281/zenodo.18841259). The structural connectivity matrices (ReDLat and non-video game players) and the source-localized EEG time series are also available in the same repository. Simultaneous EEG-fMRI data is openly available in an already published work [https://doi.org/10.1016/j.neuron.2014.03.020], although the parcellated time-series can also be found at https://zenodo.org/records/16755776 [https://doi.org/10.1371/journal.pcbi.1012852]. We used the BrainNet Viewer toolbox [https://doi.org/10.1371/journal.pone.0068910] and FSL-FMRIB [https://doi.org/10.1016/j.neuroimage.2011.09.015] for visualization.

**Funding:** AI is supported by grants from the Multi-partner consortium to expand dementia research in Latin America [ReDLat, supported by Fogarty International Center (FIC), National Institutes of Health, National Institutes of Aging (R01 AG057234, R01 AG075775, R01 AG21051, R01 AG083799, CARDS-NIH, R01 AG057234), Alzheimer's Association (SG-20-725707), Rainwater Charitable Foundation – The Bluefield project to cure FTD, and Global Brain Health Institute)], ANID/FONDECYT Regular (1250091 and 1210176 and 1220995); ANID/PIA/ANILLOS ACT210096; JPI JPND-Care, DISCeRN 2025 - Health and Social Care Research with a Focus on the Moderate and Late Stages of Neurodegenerative Diseases; FONDEF ID20I10152, and ANID/FONDAP 15150012; Wellcome Trust award for BRAIN-CLIMA: Investigating the Combined Impact of Heat and Air Pollution on Blood-Brain Barrier Integrity and Brain Aging in Latin America, (335293/Z/25/Z), and the CliCBrain (Horizon ID: 101236426; DOI 10.3030/101236426, Marie Skłodowska-Curie Actions - MSCA). This research was also supported by the National Science Centre (Poland) (2013/11/N/HS6/01335 to NK-G). RH was partially supported by the Ramón y Cajal Fellowship (RYC2022-035106-I to RH) from the Spanish Ministry of Science and Innovation/Agencia Estatal de Investigación (AEI) and the European Social Fund (FSE), and by the María de Maeztu

coupling, high firing rates, and a high proportion of γ versus α subpopulations. The ISP was necessary in the multi-frequency model to successfully fit EEG functional connectivity across frequency bands. Finally, ISP-controlled reductions in excitability reproduced both the slow-wave activity and the reduced connectivity in NREM sleep. Altogether, our model is compatible with biological evidence of the effects of excitability on modulating brain rhythms and connectivity, as observed in sleep, neurodegeneration, and chemical neuromodulation. This biophysical model with ISP provides a springboard for realistic brain simulations in health and disease.

## Author summary

Macroscale brain activity can be captured using techniques like EEG and fMRI. However, the granular or more detailed activity of neurons and localized neural masses is inaccessible. A solution is the use of whole-brain models, as they can simulate EEG and fMRI recordings from mathematical equations and can be fit to empirical data. One limitation in these models is hyperexcitability (over-saturation, or runaway excitation). When the coupling between brain areas increases, they might become aberrantly hyperexcitable if no compensatory mechanisms are considered. To address this, our framework includes a mechanism that dynamically modifies feedback inhibition to compensate for this excitability increase during simulations. Here, we systematically characterize and validate a previously introduced model with inhibitory synaptic plasticity, a mechanism that can directly control brain excitability and keep it stable in brain simulations. We ran different types of simulations and analyses to fully characterize the model. We observed that controlling system excitability is necessary to fully capture EEG connectivity and to simultaneously reproduce the EEG power spectrum and fMRI connectivity. Moreover, we showed that reduced/increased brain excitability is associated with the emergence of the slow/fast EEG rhythms. The model can be used to characterize how connectivity and brain dynamics are altered in different types of conditions, such as chemical neuromodulation, drug delivery, altered states of consciousness, and neurodegenerative disorders. Our model is open access, well-documented, and accompanied by tutorials to make it accessible to the whole neuroscience community.

## 1. Introduction

Understanding the biophysical mechanisms that give rise to large-scale brain dynamics remains a central topic in neuroscience [1,2]. Generative models of brain activity offer a principled approach to bridge empirical observations and underlying mechanisms [3–5]. These models are mathematical tools used to simulate neural population dynamics and are constrained by anatomical and neurophysiological priors. These models are particularly useful in capturing macro-scale patterns observed in

Program for Units of Excellence in R&D (CEX2021-001164-M/10.13039/501100011033 to RH). The contents of this publication are solely the authors' responsibility and do not necessarily represent the official views of these institutions. The funding agencies had no role in study design, data collection and analysis, decision to publish, or preparation of the manuscript.

functional magnetic resonance imaging (fMRI) and electroencephalography (EEG) [2,6–8]. A key challenge, however, lies in setting these models to reflect biologically plausible excitation/inhibition (E/I) balance, especially in whole-brain simulations where runaway excitation (hereafter 'hyperexcitability' or over-saturation) can dislocate realistic network behavior [6,9,10]. This over-saturation, caused by excessive excitatory inputs to neuronal populations, can saturate the model input-output function, disrupting oscillatory activity in biophysical models. Here, we study and validate a previously introduced multi-frequency whole-brain model [7,11], from the Jansen-Rit equations [12,13], with an inhibitory synaptic plasticity (ISP) mechanism [7,9,10]. We characterize its dynamical repertoire, parameter dependence, and empirical fitting properties. Our approach allows simultaneous fitting of EEG and fMRI empirical data, providing a mechanistic account of neurophysiological changes during different brain states, e.g., wakefulness and NREM sleep.

Whole-brain models simulate the interactions of distributed brain regions and are commonly employed to explore how anatomical connectivity, neuromodulation, and local dynamics shape brain activity [2,3,5,14]. These models vary in their level of abstraction; they range from pure abstract phenomenological models [15–17], to biophysically grounded models that aim to reproduce underlying neural principles [6,18–20]. The latter can be used to propose biophysical mechanisms in healthy and pathological brain dynamics [3,7,8,21–24], making them a particularly valuable tool in translational neuroscience. When constrained by empirical data, these models facilitate the identification of biomarkers, and provide a framework for evaluating the potential effects of neuromodulation or pharmacological interventions in silico [25,26]. Biophysical models can also serve as testbeds for perturbational strategies aimed at restoring pathological brain dynamics [27].

Classical neural mass models, such as the Wilson-Cowan or Jansen-Rit, remain some of the workhorses for simulating large-scale brain dynamics, in the form of whole-brain models [5]. These models undergo bifurcations, such as Hopf bifurcations [28,29], beyond which node dynamics converge to stable fixed points and cease to oscillate. This transition disrupts coherent oscillations and prevents the emergence of globally correlated network activity. Both the classical Jansen-Rit model [12], as well as multi-frequency extensions [30], can enter hyperexcitable or over-saturated regimes in whole-brain simulations. Here, we use the term "model hyperexcitability" to refer to this behavior: when the bifurcation occurs with respect to an external input parameter, or specifically, when it causes the over-saturation of the input-output function of the neural mass models. This model hyperexcitability is particularly observed in large-scale networks with increased inter-regional coupling, where excessive effective excitation drives the neural mass input-output sigmoid into saturation, thereby collapsing oscillations toward a fixed point [6,9,10,31]. This should not be conflated with biological/clinical hyperexcitability, which denotes a pathophysiological increase in neuronal excitability and/or E/I imbalance in vivo.

Previous works have suggested that homeostatic inhibitory synaptic plasticity (ISP) can counteract model hyperexcitability by implementing activity-dependent feedback inhibition. The inhibitory strength is adjusted in an on-line manner to stabilize firing rates (and thus regional operating points), preventing saturation and helping

to maintain sustained oscillations and network-wide integration [7,9,10]. The conditions under which a given whole-brain model enters this saturated, non-oscillatory regime are model- and fitting-dependent. One advantage of the ISP dynamical mechanism, compared to heuristic approaches [21,31], is that it does not require a priori knowledge of how the model's many interacting parameters jointly modulate excitability. Rather than relying on pre-defined parameter tuning, the feedback inhibition is adjusted "on-the-run" based on ongoing activity, allowing firing rates to be clamped with relative robustness to changes in the rest of the parameters.

Here, we study a previously introduced multi-frequency neural mass model used in dementia [7,30], based on the Jansen-Rit equations [12,30], that includes a biologically motivated ISP mechanism [9,10]. We begin by analyzing the local effects of ISP through bifurcation analyses of single-node dynamics, demonstrating how inhibitory control modulates the transition between oscillatory and non-oscillatory states. We then explore how key model parameters influence the spectral content of simulated EEG signals. Using source-reconstructed EEG data from healthy participants, we fit the model to reproduce empirically observed functional connectivity (FC) across multiple frequency bands. Finally, we extend the model to simultaneously fit both EEG power spectra and fMRI FC during wakefulness and NREM sleep by modulating the E/I balance of neural masses. Through this work, we provide a generative framework, grounded in physiological mechanisms, that accounts for both spectral and connectivity features of brain dynamics.

## 2. Results

We used a modified version of the Jansen-Rit model (Fig 1) [7,11]. The model combines empirical priors (Fig 1A) with a dynamical model for simulating neural activity from cortical columns (Fig 1B). Here, we provide a full characterization of the model from the single-node to the whole-brain dynamics. At the nodal level, our model consisted of two subpopulations of cortical columns (Fig 1B). The parameters of these subpopulations defined their intrinsic oscillatory frequencies. Specifically, the neural gains for excitatory, $A$, and inhibitory, $B$, populations, and the inverse of characteristic time constants for excitatory, $a$, and inhibitory, $b$, synapses were chosen to control the frequency of oscillations (see Table 2 with the model's parameters). The parameters allow subpopulations to oscillate within the α or γ EEG canonical frequency bands [30]. The proportion of α versus γ neurons, $r^a$, controls the contributions of each subpopulation to the EEG-like activity [7,30], determining the spectral properties of individual brain areas. The other element of our model is ISP (Fig 1B), modeled as an additional differential equation for adjusting the feedback inhibition onto pyramidal neurons, $C_4$, to dynamically set the firing rate of pyramidal neurons around a target value, $\rho$. A synaptic plasticity time constant, $\tau$, controls the speed of convergence to the target firing rate value.

### 2.1 Analysis of Jansen-Rit single-node dynamics

We characterized the effect of plasticity by studying the model at the single-node level, considering only the α population ($r^a = 1$). Without plasticity, this corresponds to the original Jansen-Rit model in its most common implementation and parameter set (with a slight modification of neural gains and time constants, as presented in Table 2). Figure 2A shows a bifurcation diagram of this model against the $p$ parameter (external input), showing the stability of the fixed point $x_0$ (excitatory inputs to pyramidal neurons), the frequency of the detected oscillations, and two examples of output voltage. As it has been previously described [29,34], the model presents robust oscillations of ~10 Hz within a suitable range of $p$.

The addition of ISP into the model dramatically reshapes the bifurcation diagram of the model (Fig 2B, 2C). Now, oscillations are sustained robustly in a much wider range of $p$ (note the difference of scale in the $X$ axes of Fig 2B vs 2A). The frequency of oscillation is not only dependent on the input $p$, but also on the target value $\rho$ for the firing rate of the pyramidal population. Here, $\rho$ denotes a homeostatic target value for the time-averaged pyramidal firing rate (regulated by ISP) and should not be interpreted as a target oscillation frequency; instead, changing $\rho$ shifts the E/I operating point via adaptive inhibition, thereby moving the system across dynamical regimes with different emergent oscillation frequencies. When $\rho$ is set to 3.5 Hz, oscillations are set very robustly around the α band (10 Hz) (Fig 2C). Also, with $p = 220$ Hz, the modulation of amplitude resembles the behavior of the original model at the same input value [28,29].

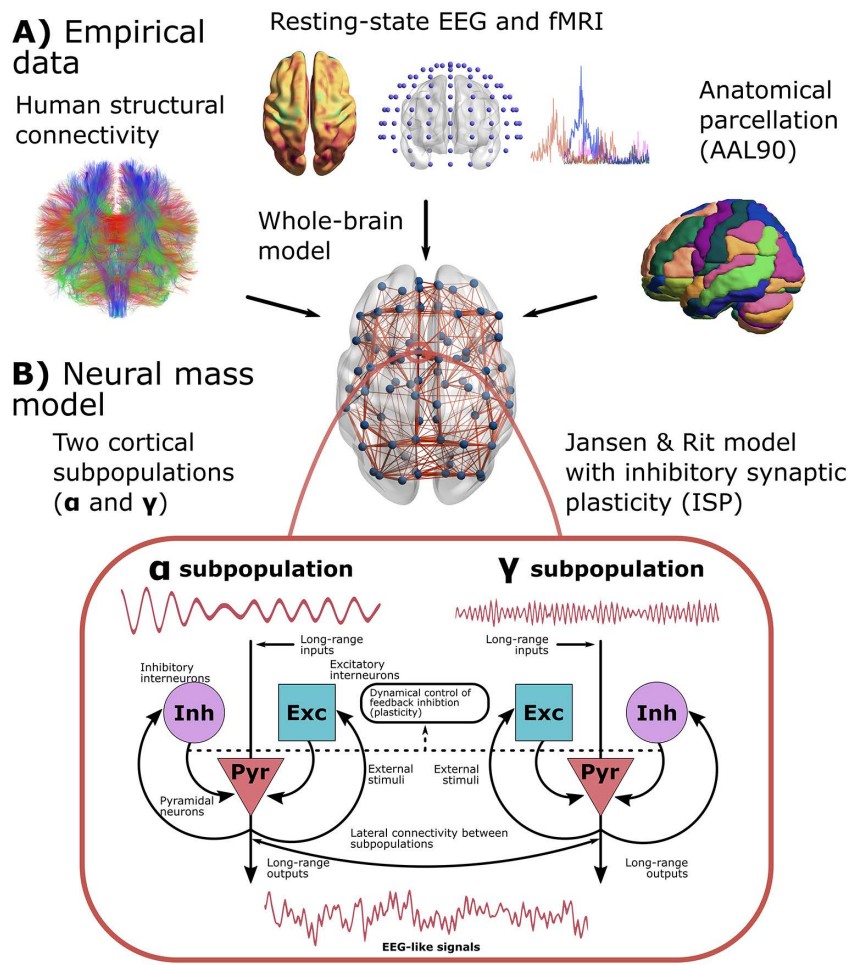

**A)** Empirical data

Human structural connectivity

Resting-state EEG and fMRI

Whole-brain model

Anatomical parcellation (AAL90)

**B)** Neural mass model

Two cortical subpopulations (**α** and **γ**)

Jansen & Rit model with inhibitory synaptic plasticity (ISP)

**α subpopulation**          **γ subpopulation**

Inhibitory interneurons

Long-range inputs

Excitatory interneurons

**Inh**   **Exc**

Dynamical control of feedback inhibtion (plasticity)

Long-range inputs

**Exc**   **Inh**

**Pyr**

Pyramidal neurons

External stimuli   External stimuli

**Pyr**

Lateral connectivity between subpopulations

Long-range outputs

Long-range outputs

**EEG-like signals**

**Fig 1. Whole-brain neural mass model. A)** The model is constrained with structural connectivity (SC, obtained from DTI), fMRI/EEG functional connectivity (FC), and the EEG power spectrum. Brain areas were parcellated using the AAL90 brain parcellation. **B)** Each brain area in the whole-brain model consisted of a modified version of the Jansen-Rit neural mass model. Each region is modeled using two subpopulations of neural masses with a frequency peak of oscillation within the α and γ bands of the EEG spectrum. Both subpopulations include the interactions between pyramidal and interneuron populations. The model also included inhibitory synaptic plasticity (ISP), as an additional equation to model the time courses of the feedback inhibition onto pyramidal populations, to reach a desired average firing rate. The brain plots in (A) and (B) were generated using BrainNet Viewer for MATLAB [32]. The DTI-based network in (A) was generated using FSL-FMRIB [33].

Across the bifurcation diagrams in Fig 2, oscillatory stability is shaped by bifurcation of the limit cycle rather than by a saddle-node on invariant circle mechanism. In Fig 2A, the stability change of the limit cycle around $p \approx 140$ corresponds to a fold of cycles (saddle-node of limit cycles). In Fig 2B, between the saddle-node bifurcation at $p = 6.62$ and the torus bifurcation at $p = 111$, deterministic simulations do not converge to a stable attractor and instead display persistent oscillatory switching between two regions of state space (around $x_0 \approx 0.005$ and $x_0 \approx 0.15$). Because this regime is not central to the aims of the present work, we only report its consequences in S1 Fig. In Fig 2C, the system always exhibits a stable attractor (one or two stable fixed points or a stable limit cycle) for the explored input range, and no divergent behavior is observed. We also analyzed bifurcations with respect to the target firing rate ρ (S2 Fig). As in [10], varying ρ shifts the E/I operating point and can drive transitions between non-oscillatory (fixed-point) and oscillatory regimes, typically via a Hopf

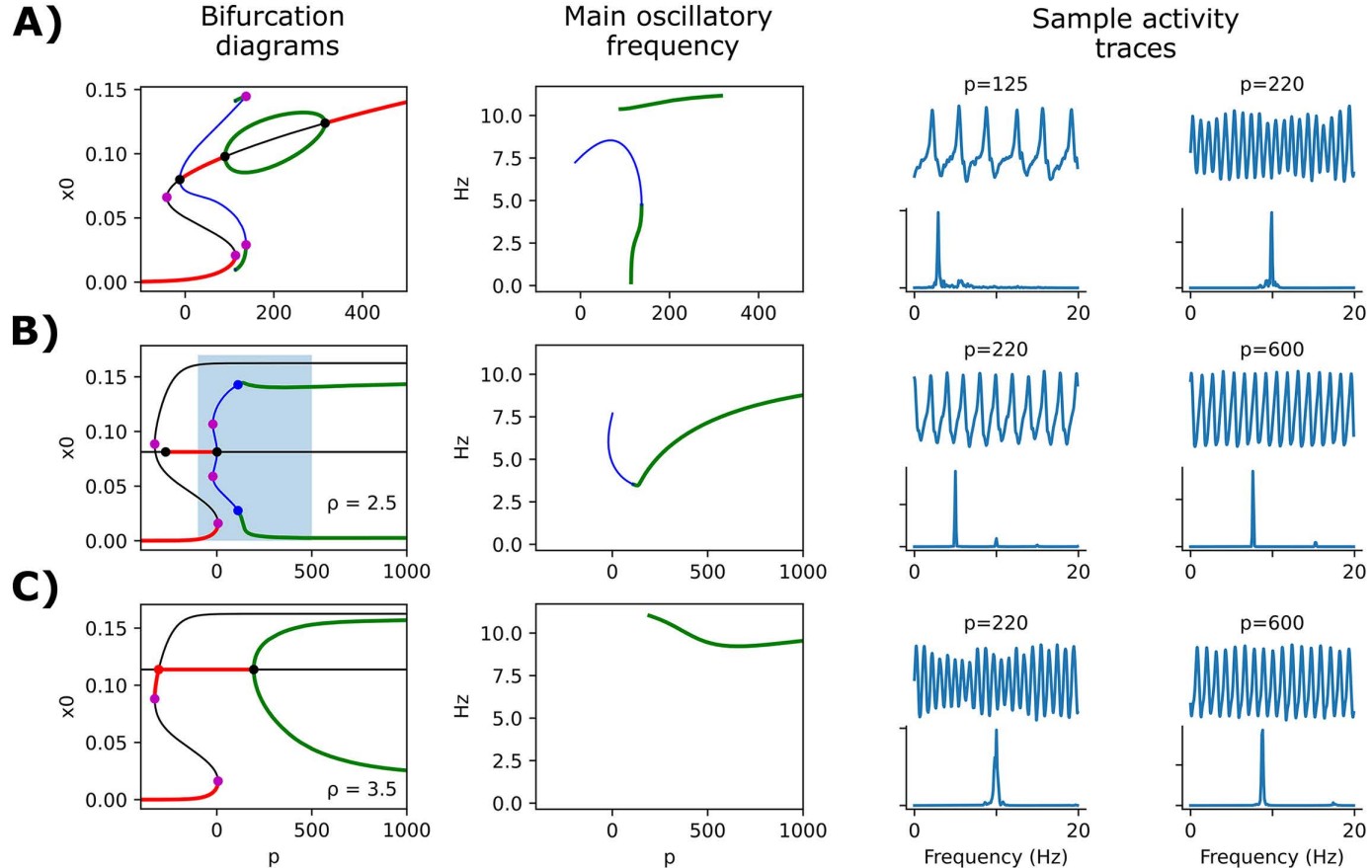

**Fig 2. Bifurcation analysis of the model with and without plasticity. A)** Bifurcation diagram, main oscillatory frequency, and two sample activity traces at the indicated $p$ input values, for the original Jansen-Rit model without plasticity and consisting only of α population (10 Hz). Below the traces, a power spectral density (PSD) plot is shown. In the left panel, red thick lines and black lines represent stable and unstable fixed points, respectively. Colored lines represent maxima and minima of stable (green) and unstable (blue) periodic orbits. The colors are maintained in the center panel to show the frequency (in Hz) of the associated oscillations. Dots denote the bifurcations encountered: purple = saddle-node bifurcations and saddle-node of limit cycles; black = Hopf bifurcation. Simulations include a noise factor added to the $p$ parameter. **B)** Same as in **(A)**, with the addition of the homeostatic plasticity mechanism and a target for pyramidal target firing rate $\rho = 2.5$ Hz. Blue dots denote a Torus or Neimar-Sacker bifurcation. The shaded rectangle in the left panel denotes the range of $p$ that is plotted in **(A)**. **C)** Same as in **(B)**, with the target $\rho = 3.5$ Hz. Red dot denotes a branching or pitchfork bifurcation.

bifurcation (along with additional bifurcations in some parameter ranges). Consistent with this, $\rho$ also modulates the oscillation frequency, with higher frequencies generally obtained at larger $\rho$ values.

Then, we analyzed the effect of combining the two cortical subpopulations, α and γ, on nodal oscillatory frequency. We ran simulations without the ISP to characterize the EEG power spectrum shape (Fig 3A). Using single-node simulations, we modified the $r^a$ parameter to control the contribution of α versus γ neurons within the cortical columns. For $r^a < 0.25$, the single node oscillates within the γ EEG frequency band (> 30 Hz). On the other hand, $r^a > 0.9$ produces mainly α oscillations (with a frequency peak around 10 Hz). For intermediate values of $r^a$, the model exhibits a richer power spectrum, with distributed power around all the possible canonical frequency bands. Using the multi-frequency model, we ran single-node simulations with $r^a = 0.5$ to check the model's capability to clamp firing rates to target values. Across different values of external stimulation, $p$, and target firing rates, $\rho$, the model adjusts the feedback inhibition (the inhibitory-to-excitatory local connectivity constant $C_4$) to reduce/increase excitability and finally match the desired firing rates $\rho$ (Fig 3B, 3C).

PLOS Computational Biology

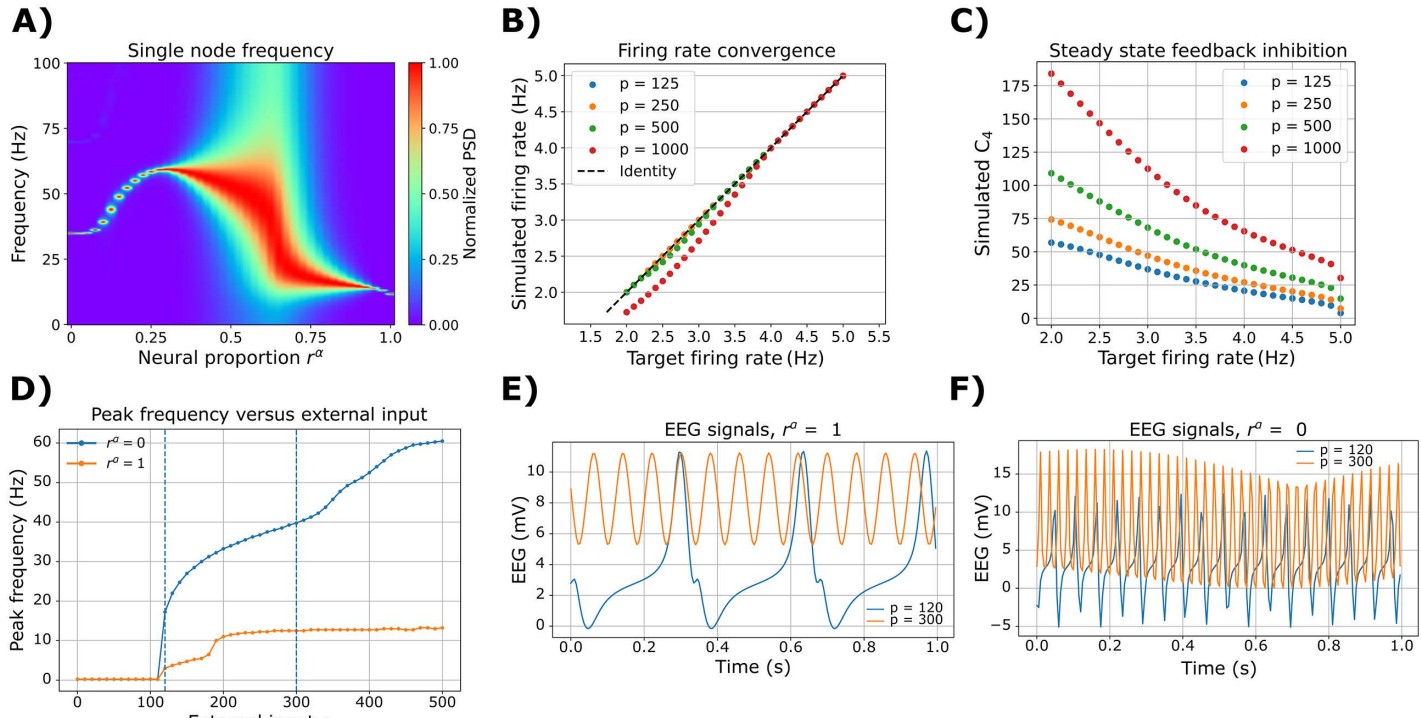

**Fig 3. Single-node dynamics. A)** Modulation of oscillatory frequency in single-node simulations. Normalized PSD as a function of α versus γ subpopulation proportion, $r^\alpha$, in the single-region model. The values shown here are the average of 50 random seeds. Note that PSD estimates are computed on a discrete frequency grid (Welch), so narrow-band ridges may appear step-like with $r^\alpha$. **B)** Noise-free simulations with fixed $r^\alpha = 0.5$, showing the target firing rate $\rho$ versus the time-averaged pyramidal firing rate produced by the model. The ISP is "clamping" the ⟨rate⟩ ≈ $\rho$, which is an indicator of a successful feedback inhibition regulation. **C)** Same simulations as in (B), showing the target firing rate $\rho$ versus the time-averaged feedback inhibition, given by the inhibitory-to-excitatory local connectivity parameter $C_4$ (the steady-state value reached by ISP to achieve the clamping in B). **D)** Peak frequency versus external input for the "fast" γ subpopulation ($r^\alpha = 0$) and the "slow" α subpopulation ($r^\alpha = 1$). The dashed vertical lines correspond to the traces in **E)** for $r^\alpha = 1$, and **F)** for $r^\alpha = 0$.

Finally, we examined the "slow" limit-cycle frequencies for each subpopulation, in addition to the fast α/γ regimes. For both the α ($r^\alpha = 1$) and γ ($r^\alpha = 0$) subpopulations, the model exhibits two oscillatory regimes depending on external input $p$ (Fig 3D–3F). At lower input ($p = 120$; slow regime), the α subpopulation shows the expected low-frequency-limit cycle with a peak around ~3 Hz (δ/θ range), consistent with the classical Jansen-Rit slow cycle. The γ subpopulation also presents a slow limit cycle, but with a higher peak frequency around ~20 Hz (β range). At higher input ($p = 300$; fast regime), the α and γ subpopulations express their respective fast limit cycles (α ~ 10 Hz; γ > 30 Hz), indicating that the γ tuning shifts both fast and slow oscillatory regimes while preserving the presence of a distinct slow cycle.

## 2.2 Whole-brain multi-frequency model with ISP

Here, we investigated the behavior of the multi-frequency model with ISP when many regions were coupled. In this setting, the ISP mechanisms were used to prevent hyperexcitability. Based on the results of the previous section, we first fixed the target firing rate $\rho$ to 2.5 Hz and $r^\alpha = 0.5$ [7,35] (Fig 4), as they generated robust oscillatory activity in a wide spectrum of frequencies. To connect different regions, we employed structural connectivity (SC) data of 45 healthy controls from the ReDLat consortium [36]. Data were parcellated into 90 regions using the AAL90 brain atlas [37] (Table 1), excluding all subcortical areas except the amygdala and hippocampus. We swept the global coupling parameter, $K$, from no coupling

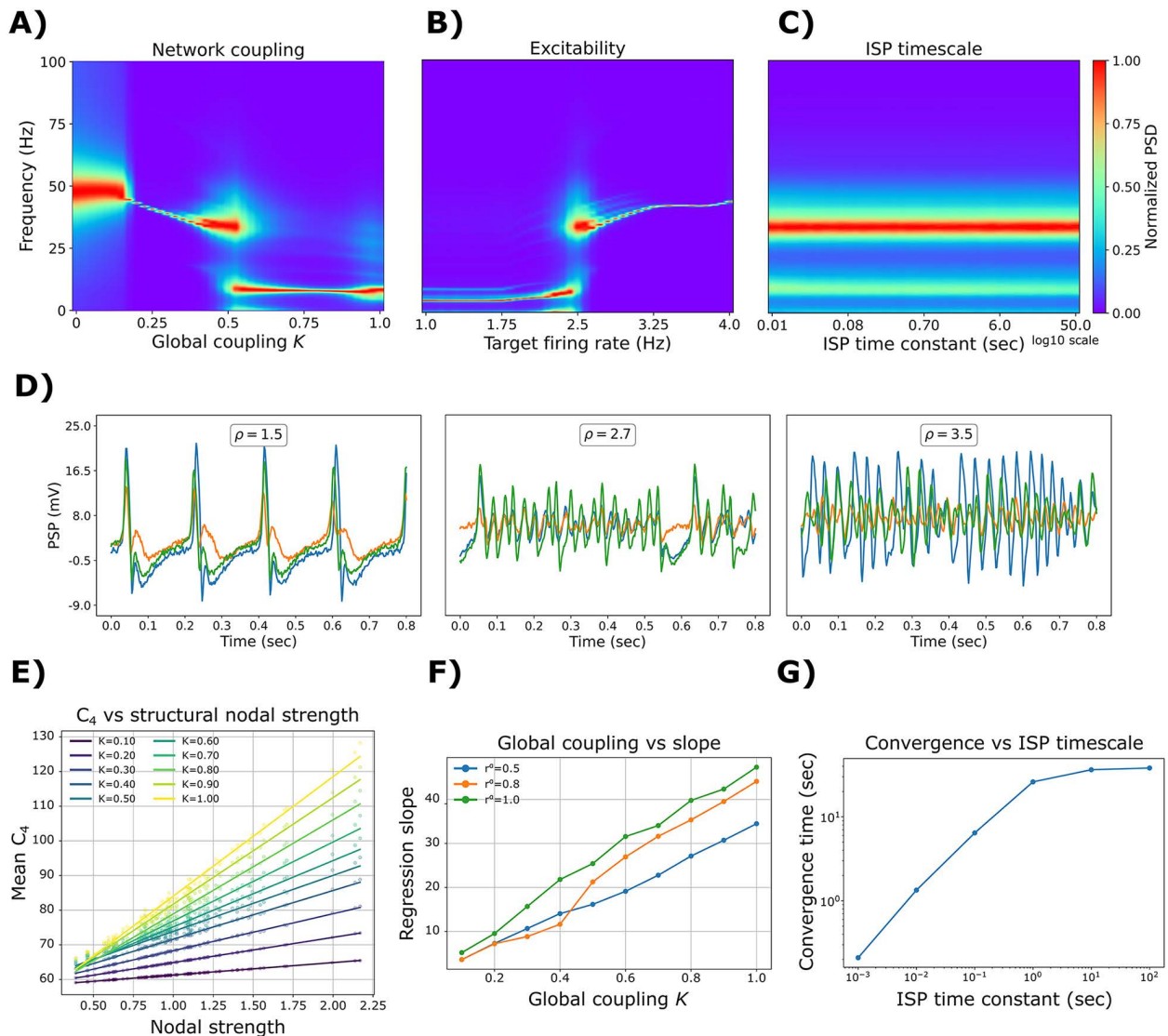

**Fig 4. Modulation of oscillatory frequency in whole-brain simulations.** Normalized power spectral density (PSD) as a function of: **A)** global coupling, $K$, in the whole-brain model, with a fixed target of $\rho=2.5$ Hz. **B)** target (desired) firing rate in the whole-brain model, with a fixed $K=0.5$. **C)** inhibitory synaptic plasticity (ISP) time constant, $\tau$. The values shown here are the average of 50 random seeds. **D)** Example of EEG traces for a fixed $K=0.5$, $r^\alpha=0.5$, $\tau=2$ sec, and different values of $\rho$. **E)** Relationship between the nodal strength, from the structural connectivity matrix, and the time-averaged feedback inhibition (mean $C_4$), for different values of global coupling $K$. **F)** Slopes of the linear regressions in (E), as a function of global coupling. 10 random seeds were used for these simulations. **G)** Convergence time vs ISP timescale. The convergence time (in seconds) was defined as the first time the ROI-averaged C4 reached 63.2% of its total change from the initial to the final steady state in a noise-free simulation.

($K=0$) to strong coupling ($K=1$) (Fig 4A). For $K=0$, the local dynamics are constrained to solely γ oscillations. Increasing the global coupling to $K=1$ generates α and θ rhythms. In the middle, with $K=0.5$, a region of coexistence of faster (β, γ) and slower (θ, α) oscillations emerged at the whole-brain level. The spread of the frequency of oscillation, for intermediate values of $r^\alpha$, might be a consequence of the small stochastic fluctuations in the external input, to mimic background variability in afferent drive. This does not alter the underlying oscillatory regime, but it broadens PSD peaks and can produce a small apparent spread in the dominant frequency estimated from finite-length time series, especially near transitions.

**Table 1. List of brain regions of AAL90 parcellation.** The atlas comprises 90 cortical and subcortical brain areas (45 per hemisphere). Regions marked by * were included for fMRI simulations, but not for EEG.

**Brain Regions**

| Name | Abbreviation | Name | Abbreviation |
|---|---|---|---|
| Precentral gyrus | PreCG | Lingual gyrus | LING |
| Superior frontal gyrus (dorsolateral) | SFGdor | Superior occipital gyrus | SOG |
| Superior frontal gyrus (orbital) | ORBsup | Middle occipital gyrus | MOG |
| Middle frontal gyrus | MFG | Inferior occipital gyrus | IOG |
| Middle frontal gyrus (orbital) | ORBmid | Fusiform gyrus | FFG |
| Inferior frontal gyrus (opercular) | IFGoperc | Postcentral gyrus | PoCG |
| Inferior frontal gyrus (triangular) | IFGtriang | Superior parietal gyrus | SPG |
| Inferior frontal gyrus (orbital) | ORBinf | Inferior parietal gyrus | IPG |
| Rolandic operculum | ROL | Supramarginal gyrus | SMG |
| Supplementary motor area | SMA | Angular gyrus | ANG |
| Olfactory cortex | OLF | Precuneus | PCUN |
| Superior frontal gyrus (medial) | SFGmed | Paracentral lobule* | PCL |
| Superior frontal gyrus (medial orbital) | ORBsupmed | Caudate* | CAU |
| Rectus gyrus | REC | Putamen* | PUT |
| Insula | INS | Pallidum* | PAL |
| Anterior cingulate gyrus | ACG | Thalamus* | THA |
| Median cingulate gyrus | MCG | Heschl gyrus | HES |
| Posterior cingulate gyrus | PCG | Superior temporal gyrus | STG |
| Hippocampus | HIP | Temporal pole (superior) | TPOsup |
| Parahippocampal gyrus | PHG | Middle temporal gyrus | MTG |
| Amygdala | AMYG | Temporal pole (middle) | TPOmid |
| Calcarine cortex | CAL | Inferior temporal gyrus | ITG |
| Cuneus | CUN | | |

When fixing global coupling to $K = 0.5$, $r^{\alpha} = 0.5$, and then manipulating target firing rates, $\rho$, we can shift brain dynamics from slower (for $\rho < 2.5$ Hz) to faster (for $\rho > 2.5$ Hz) regimes of activity (Fig 4B). Remarkably, we found no effect of the ISP time constant, $\tau$, on the oscillatory frequency of the model (Fig 4C). Example EEG traces are depicted in Fig 4D, for $K = 0.5$, and $r^{\alpha} = 0.5$. There, low $\rho$ values produce δ-θ oscillations, intermediate values α rhythms, and the highest values produce faster β-γ activity.

Across whole-brain simulations, we examined how the ISP shapes the spatial distribution of feedback inhibition and how the ISP timescale affects the adaptation dynamics (Fig 4E–4G). For each value of global coupling $K$, the time-averaged feedback inhibition (mean $C_4$) was computed and related to the nodal strength of the structural connectivity matrix (Fig 4E). Mean $C_4$ increased with nodal strength, and this relationship was well approximated by a linear regression for each $K$ value. Then the slope of this regression was expressed as a function of $K$ (Fig 4F), showing that feedback inhibition changes systematically with global coupling. Importantly, while this mapping could in principle be used to pre-compute a set of node-specific $C_4$ values for a given $K$, the regression slope (and thus the inferred $C_4$ distribution) also depends on other model parameters (e.g., the α/γ mixture $r^{\alpha}$), limiting the generalizability of any fixed pre-computed inhibition across the full parameter space. To assess the role of the ISP time constant, we quantified the convergence time of the plasticity variable in noise-free simulations (Fig 4G). Convergence timescale (seconds) was defined as the first time at which the ROI-averaged $C_4$ reached 63.2% of its total change from the initial value to its final steady-state

value. Across the tested time constant values $\tau$, this parameter primarily modulated the convergence timescale, while the steady-state oscillatory regime (and the normalized PSD, Fig 4C) remained largely unchanged once the system had converged.

In summary, slower oscillatory regimes emerge from a high proportion of $\alpha$ over $\gamma$ neurons ($r^a$), high global coupling $K$, and low $\rho$, whereas faster oscillations appear in the opposite direction. In addition, ISP shapes a node-specific inhibition profile that scales with nodal strength in a $K$-dependent manner, while the ISP timescale primarily controls the convergence time of $C_4$ to its steady state.

## 2.3 Fitting to EEG source functional connectivity data

To further validate our model, we assessed its capacity to fit FC across the whole EEG power spectrum (Fig 5). We used the same SC data [36] and brain parcellation [37] described in the previous section, and the source-reconstructed EEG time series from the same participants [36]. We filtered the simulated and empirical EEG signals in the canonical frequency bands ($\delta$, $\theta$, $\alpha$, and $\beta$). We calculated the signals' envelopes, which were high-pass filtered (cut-off frequency of 0.5 Hz), and FCs were computed using the amplitude envelope correlation [38]. We compared the model without (Fig 5A) and with (Fig 5B) ISP. The $r^a$ and $K$ parameters were modulated to find the best combinations to reproduce the EEG FC. Simulated and empirical matrices were contrasted using the structural similarity index (SSIM, with SSIM = 1 being a perfect fit, 0 being the opposite) [39]. We aimed to simultaneously maximize SSIM for all EEG frequency bands (maximizing the average SSIM across bands).

The model without ISP is characterized by a parameter space with a small region of correlated activity (Fig 5A). The model is unable to reach correlated activity before becoming hyperexcited, as no compensatory mechanisms were introduced to avoid that. Further, the model cannot generate correlated activity when mixing $\alpha$ and $\gamma$ subpopulations in cortical columns; that is, near 0 mean FC for $r^a$ values around 0.5 suggests model frustration (inability to generate correlated activity). This limitation reflects the local and global network dynamics of the model without ISP: for large portions of the ($K$, $r^a$) space, increased effective excitation drives the system toward over-saturation and loss of sustained oscillations, which prevents the emergence of stable band-limited envelope fluctuations and, consequently, correlated activity. Therefore, the restricted parameter region in Fig 5A appears primarily due to the model's inability to sustain oscillatory dynamics without an adaptive inhibitory mechanism, rather than a bias induced by additional fitting targets or anatomical priors. Including ISP in the model, with $\rho = 2.5$ Hz, enriches the parameter landscape with a wider range of parameters where correlations can emerge, from hypo to hyperconnectivity patterns, including intermediate connectivity regimes of activity (Fig 5B). Interestingly, correlations for intermediate values of $r^a$ are now viable. Increasing global coupling in this scenario produced multi-frequency EEG FC connectivity, with maximal SSIM values in the proximity of $r^a = 0.5$ and $K = 0.5$.

We used the optimal parameters to compare the single-frequency model (without ISP, $K = 0.425$, $r^a = 1$, SSIM = 0.14 averaged across frequency bands) and the multi-frequency model (with ISP, $K = 0.675$, $r^a = 0.675$, $\rho = 2.5$ Hz, SSIM = 0.56 averaged across frequency bands) to reproduce EEG FC (Fig 6). The SSIM and Pearson's correlation values between the two models are presented in Fig 6A. Regardless of the chosen metric to compare the goodness of fit, the multi-frequency model outperformed the classical single-frequency model across all frequency bands ($|D| > 1.2$, denoting a huge effect size) (Fig 6A). Example FCs are presented for the single and multi-frequency models, including the empirical FC matrices (Fig 6B).

To illustrate the qualitative temporal structure of the ISP-fit model beyond FC metrics, we additionally report representative EEG time series and amplitude envelopes for the best-fitting parameter set ($K = 0.675$, $r^a = 0.575$, $\rho = 2.5$ Hz) (Fig 7). We show a 10-second segment of simulated EEG, band-pass filtered into canonical frequency bands ($\delta$-$\gamma$). We also plot the $\alpha$-band envelope from the same simulation, which captures the slow $\alpha$-band power fluctuations (amplitude dynamics).

In this way, we demonstrated that ISP preserved E/I balance and enabled correlated activity without hyperexcitability. Also, the ISP mechanism combined with the multi-frequency model can generate multiband EEG FC across a wide range of parameters.

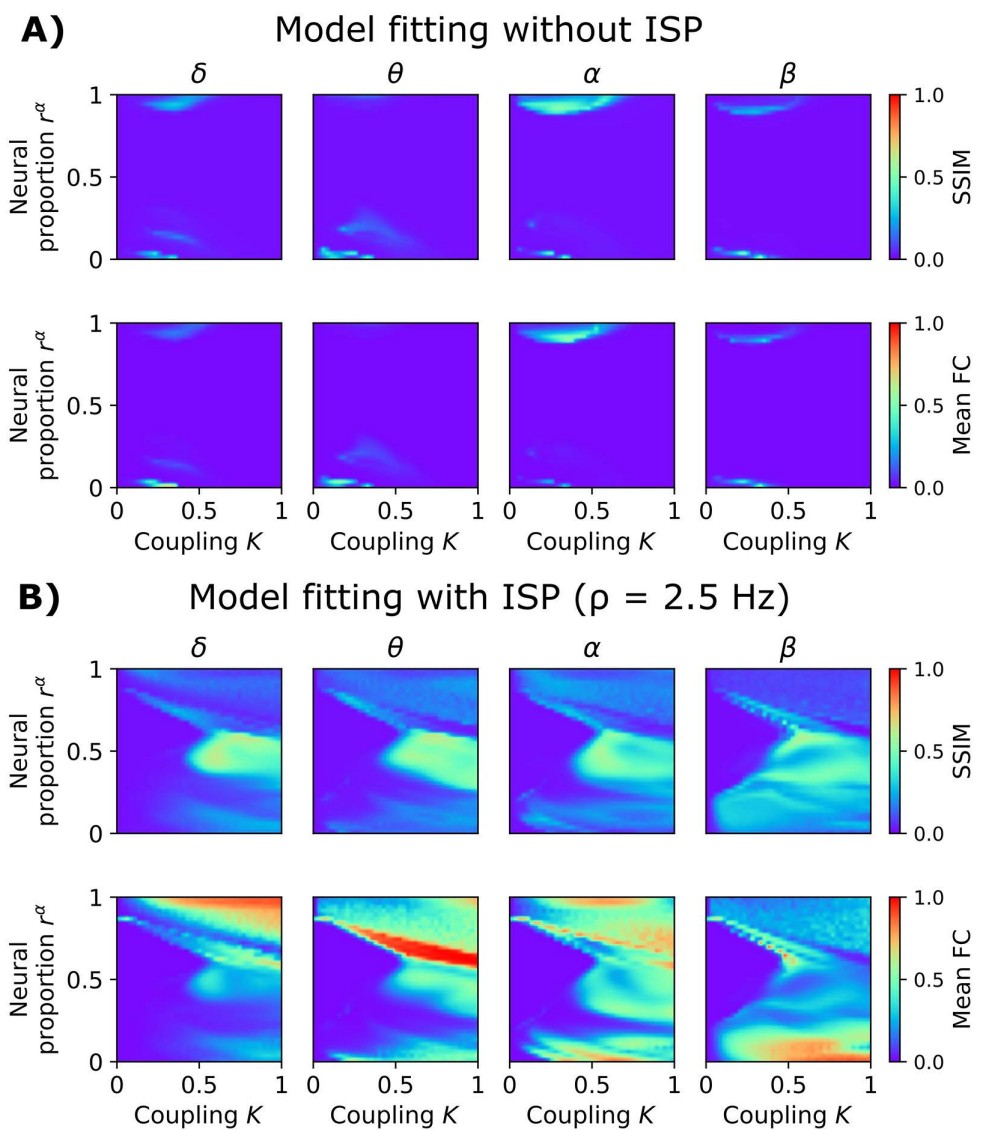

**Fig 5. Goodness of fit of the model to empirical EEG FC. A)** Model without inhibitory synaptic plasticity (ISP). The first row shows the structural similarity index (SSIM; captures the goodness of fit), and the second row represents the mean EEG FC of simulations. The axes correspond to the global coupling parameter, $K$, and the proportion of α versus γ subpopulations, $r^\alpha$. The value of SSIM = 1 indicates a good fit of the model. **B)** Goodness of fit using the model with ISP, and a fixed target firing rate, $\rho$, of 2.5 Hz. The values shown here are the average of 50 random seeds.

## 2.4 Dual fitting to EEG and fMRI

As an additional demonstration and validation of our model, we investigated its ability to jointly reproduce fMRI and EEG dynamics during NREM sleep. We used simultaneous EEG-fMRI recordings during the transition from wakefulness to different stages of NREM sleep. fMRI recordings were collected from 71 individuals, with sleep stages labeled for each fMRI volume based on simultaneously recorded polysomnographic data [40]. Here, we fitted our model to the awake (W) and N3 sleep (deep sleep). We fixed $r^\alpha = 0.5$, and systematically explored combinations of $K$ and $\rho$ drawn from a predefined parameter grid (41×41 parameter grid, with $K$ ranging from 0 to 3 and $\rho$ ranging from 2 to 3). We employed a connectome of young healthy participants from another study [41,42]. We simulated EEG-like signals and then used a hemodynamic

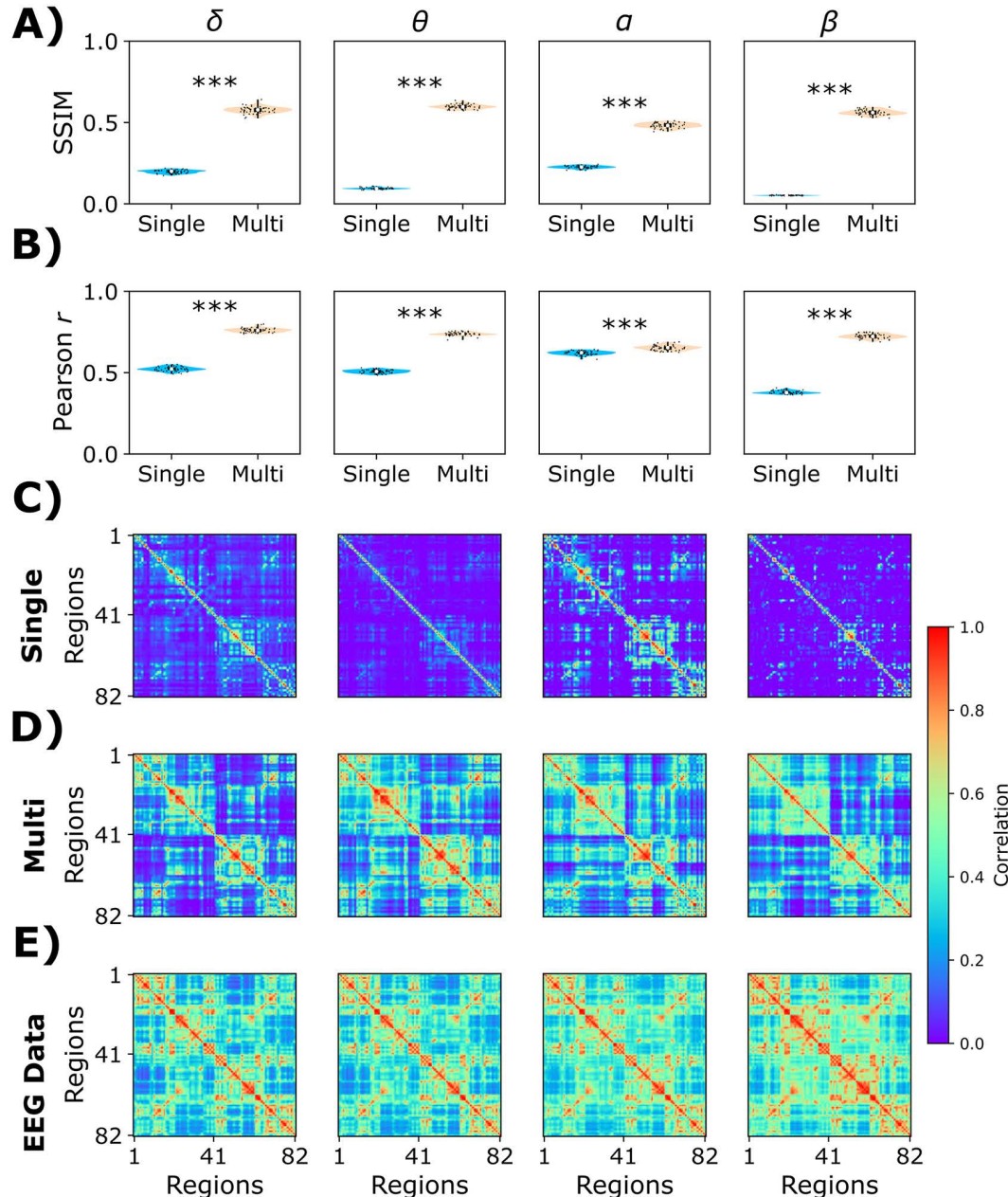

**Fig 6. Comparison of the single- and multi-frequency models in fitting empirical EEG FC. A)** The structural similarity index (SSIM) was used to measure the goodness of fit (SSIM = 1, perfect fit). **B)** Complementary to SSIM, we reported Pearson's correlation between simulated and empirical matrices (r = 1, perfect fit). The columns represent the different EEG frequency bands. **C)** Simulated EEG FCs from the single-frequency model. **D)** Simulated FCs from the multi-frequency model. **E)** Empirical EEG FC matrices. ***: |Cohen's D| > 1.2. Each point corresponds to a different model realization (50 random seeds).

model [43] to transform the firing rates into fMRI BOLD-like signals. We aimed to fit two different observables, one from each recording modality. First, the empirical averaged EEG power spectrum at the sensor level (32 EEG channels), and second, the fMRI BOLD FC (considering the AAL90 brain parcellation). From the empirical and simulated power spectrum,

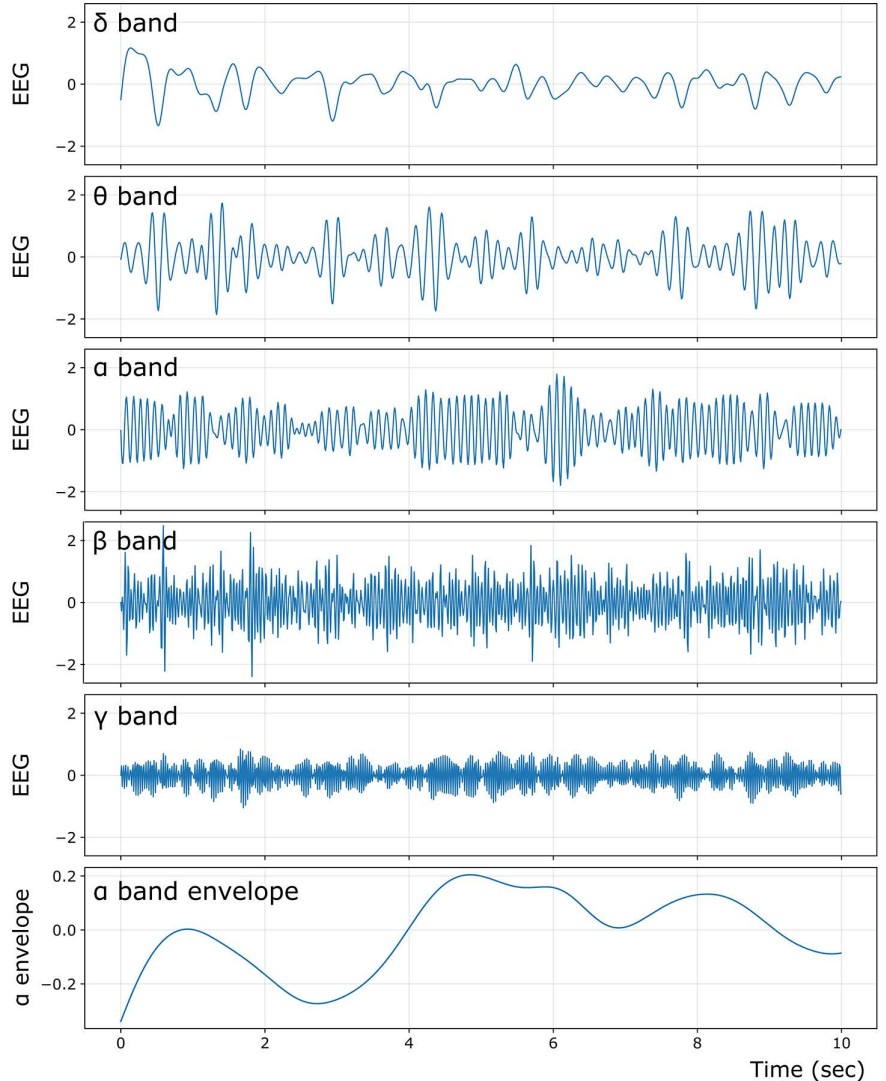

**Fig 7. Band-limited model time series and relative band power from the ISP-fit ($K=0.675$, $r^a=0.575$, $\rho=2.5$ Hz).** Representative 10-s segment of simulated EEGs, band-pass filtered into canonical frequency bands: δ (0.5-4 Hz), **θ** (4-8 Hz), α (8-13 Hz), β (13-30 Hz), and γ (30-40 Hz). The bottom panel shows the EEG α band envelope (α band power fluctuations).

we computed the θ, α, and β relative power, concatenated the values into a vector, and then compared the vectors using Clarkson's distance [44]. So, we attempted to reproduce, using whole-brain simulations, the relative contribution of θ, α, and β oscillations in EEG. For fMRI BOLD FC, we compared simulated and empirical matrices using the SSIM.

The results presented in Fig 8 suggest that using fMRI FC generates regions of the parameter space with model degeneracy in terms of goodness of fit (SSIM). That is, we observed dissimilar combinations of parameters with the same goodness of fit, regardless of whether the fitting was focused on W or N3 sleep. However, EEG fitting can be used as a mask to delimit the possible combinations of parameters that might reproduce empirical FC. In Fig 8, we used a measure of similarity (1 − Clarkson's distance) as a representation of the goodness of fit for simulated EEG, with values closer to 1 corresponding to a perfect fit to the empirical EEG relative power across frequency bands. Using a threshold of 0.85 for EEG fitting, we obtained a restricted parameter space where just a few parameters can simultaneously generate EEG and

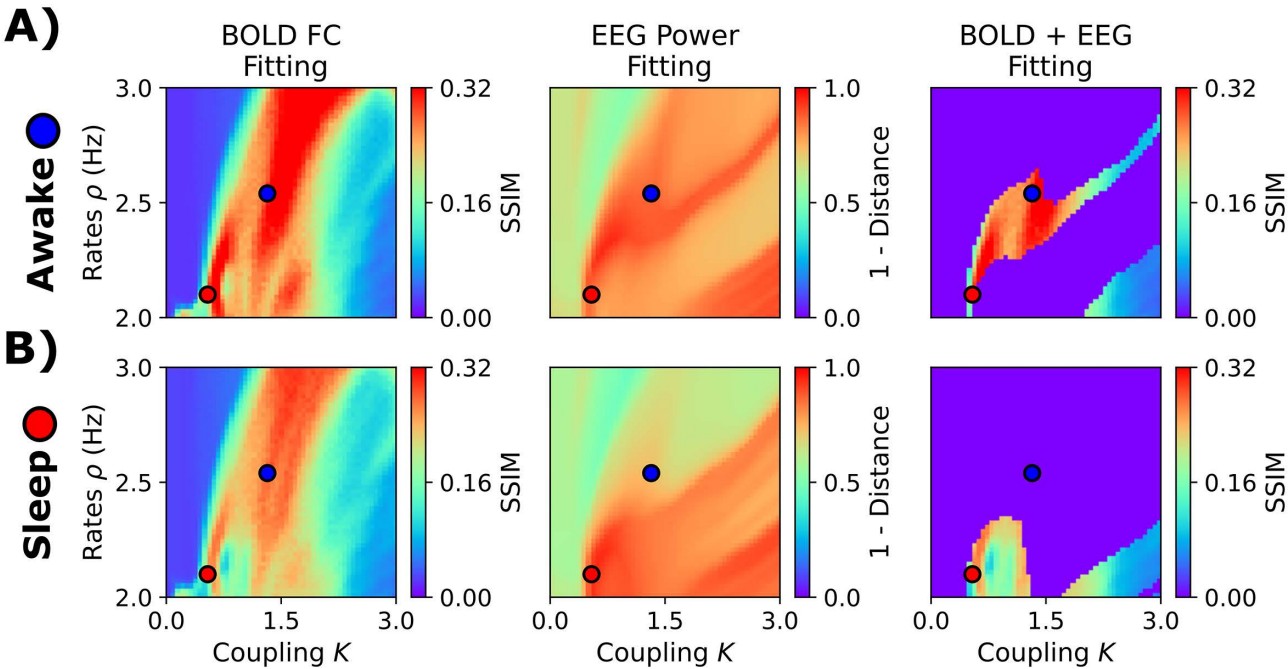

**Fig 8. Fitting the EEG and fMRI sleep dataset. A)** Fitting to awake condition. The first column corresponds to fMRI FC fitting, assessed using the structural similarity index (SSIM). The second column shows the goodness of fit to the EEG power spectrum. The last one corresponds to fMRI FC fitting using a threshold of EEG power fitting of 0.85. **B)** Fitting to EEG and fMRI NREM stage 3 (N3) data. The blue and red dots represent the best parameters (highest SSIMs when applying the EEG-fitting mask) for wakefulness ($K=1.44$, $\rho=2.56$ Hz) and N3 ($K=0.52$, $\rho=2.1$ Hz), respectively. The values shown here are the average of 50 random seeds.

fMRI dynamics that closely match empirical data. For wakefulness, the best fit was found for higher coupling and medium firing rates ($K=1.44$, $\rho=2.56$ Hz, SSIM$=0.34$, Pearson's $r=0.45$, Fig 8A). In contrast, N3 sleep was better characterized by low coupling and low firing rates ($K=0.52$, $\rho=2.1$ Hz, SSIM$=0.29$, Pearson's $r=0.38$, Fig 8B).

Simulation outcomes are displayed in Fig 9. The empirical (Fig 9A) and simulated (Fig 9B) FC matrices showed decreased connectivity strength in N3 sleep. The empirical EEG traces (example of one participant) and the group-averaged PSDs are shown in Fig 9C. Simulated EEG-like signals showed high amplitude slow oscillations for N3 compared to wakefulness (Fig 9D). Finally, the simulated power spectra presented the typical characteristics of the awake condition (peak in α) and deep sleep (power concentrated in the lowest frequency bands).

## 3. Discussion

In this work, we systematically characterized and validated a semi-empirical Jansen-Rit whole-brain model able to simulate both EEG and fMRI BOLD dynamics, from biologically plausible mechanisms. This model constitutes an improvement over previous models, combining an ISP mechanism with a multi-frequency generator of EEG-like dynamics. The ISP control loop can be used as a mechanism for testing alterations in E/I balance associated, for example, with neurodegenerative disorders or during chemical neuromodulation. When using this version of the Jansen-Rit model, we simultaneously reproduced the EEG power spectrum and fMRI FC during wakefulness and sleep. In this way, we provided the computational neuroscience community with a model for whole-brain simulations with applications in basic and clinical neuroscience.

The model extends the classical Jansen-Rit model by incorporating the ISP to modulate excitability and control the oscillatory regime of each subpopulation. In our framework, the target firing rate is a control/input parameter of the

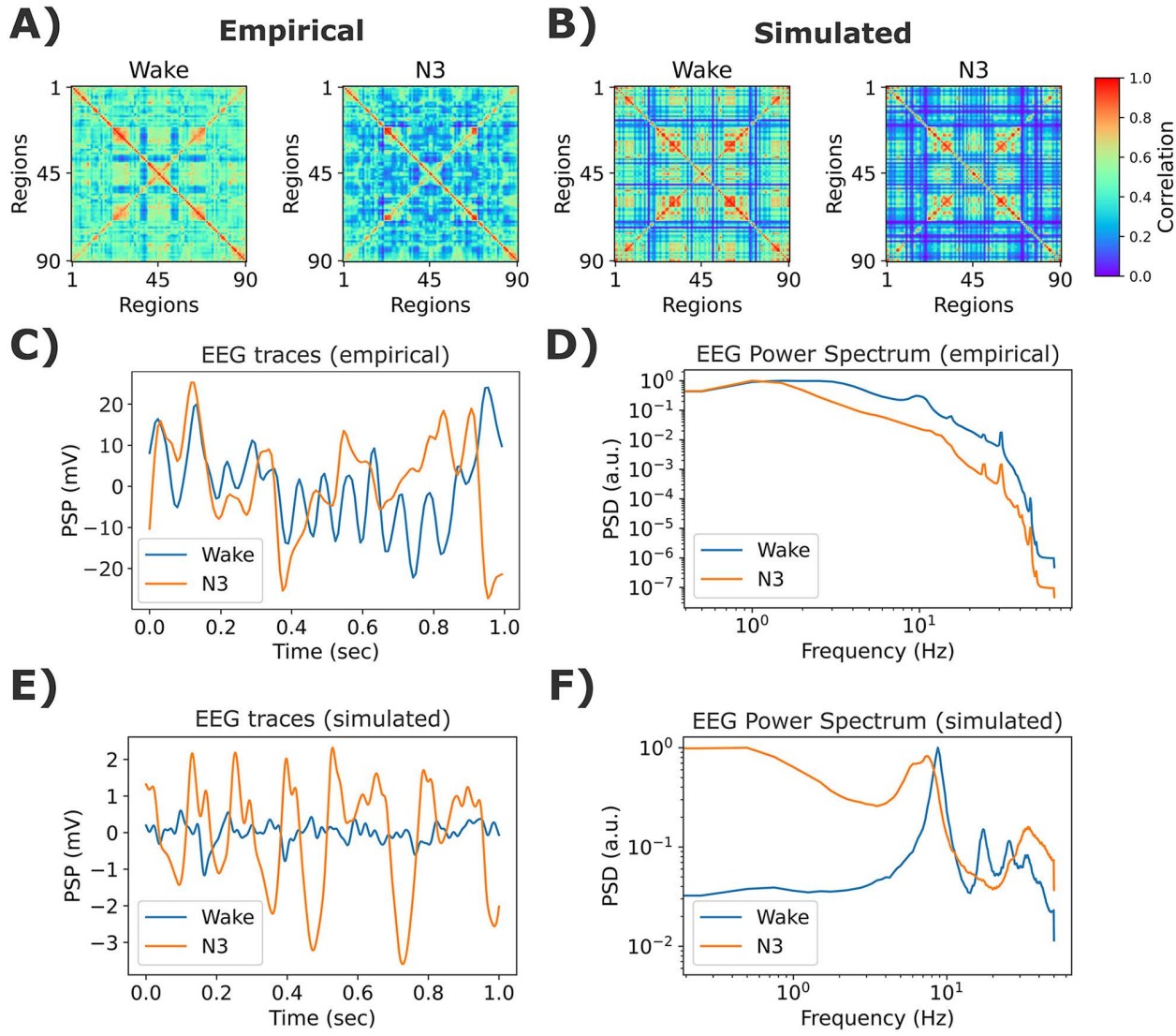

**Fig 9. Example simulations for the EEG and fMRI sleep dataset. A)** Empirical fMRI FC matrices for wakefulness and N3 sleep stages. **B)** Best simulated matrices. **C)** Empirical EEG traces (for one representative participant). **D)** Averaged empirical EEG power spectrum. **E)** Examples of EEG time courses from simulations. **F)** Simulated EEG power spectrum. The model's parameters were $K = 1.44$ and $\rho = 2.56$ Hz, for wakefulness, and $K = 0.52$, $\rho = 2.1$ Hz, for N3 sleep. The values shown here are the average of 50 random seeds.

closed-loop system (via ISP) that shifts the operating point (mean firing rate / mean inhibitory gain), thereby moving the system through parameter space [10]. In this sense, $\rho$ can move the system across bifurcation boundaries (including Hopf), although the specific boundary depends on the parameter slice being examined. The model produces slow EEG oscillations for low firing rates, in the δ and θ bands. As the firing rate increases, the system transitions toward α oscillations. In the two-subpopulation version of the model, each subpopulation has its own limit cycle, and the nodal oscillatory activity becomes a combination of α and γ rhythms. The balance between these rhythms depends on the target firing rate: higher values make γ activity more likely to be present, while lower values increase the α activity dominance. For very low firing rates, the model predominantly generates slow activity in the δ-θ range. The ISP mechanism also

provides a practical alternative to heuristic procedures that pre-estimate node-wise feedback inhibition to maintain stable whole-brain dynamics, as implemented, for example, in the Jansen-Rit [21] and Dynamic Mean Field models [31]. In these approaches, the inferred feedback inhibition is typically tied to a specific choice of coupling and local parameters and therefore becomes sensitive to changes in other model settings, making it impractical to map and fine-tune feedback inhibition across the full parameter space. In contrast, ISP adjusts inhibition online based on ongoing activity, which avoids repeated recalibration when parameters change and provides direct control of firing rates at the regional level. This allows excitability to be set explicitly for individual brain areas through a target firing rate, whereas the mapping between excitability and other model parameters is often indirect and harder to interpret.

Our model is compatible with others, showing how the modulation of excitability, driven by chemical neuromodulation [21,45,46], can produce shifts between different regimes, slower and faster, of brain dynamics. By tuning model excitability, we were able to reproduce both EEG spectral slowing and reduced fMRI FC observed during NREM sleep [45,46]. These results are consistent with established neurophysiological findings, where NREM sleep is accompanied by an overall decrease in neural excitability [47], triggered by an increased GABAergic inhibition and decreased levels of excitatory neuromodulators such as noradrenaline and acetylcholine [48,49]. In whole-brain models where NREM-like slowing is induced by reducing excitability (i.e., decreasing recurrent excitation and/or increasing global coupling), slow-wave-like activity can emerge without explicitly modeling bistable up/down states [45,46]. We recover the same qualitative mechanism: oscillatory frequency decreases when lowering the target firing rate ($\rho$) and/or increasing global coupling ($K$), consistent with our network simulations. By contrast, other Jansen-Rit whole-brain implementations modeling cholinergic/nicotinic effects have required parameter changes in the opposite direction (e.g., reduced feedback inhibition together with decreased global coupling) to reproduce an increase of cholinergic neuromodulation [21], highlighting that different neuromodulatory hypotheses can be expressed through distinct parameter pathways. Mechanistically, in our model, reduced excitability shifts the local operating point toward slower regimes by increasing effective feedback inhibition, biasing activity toward the α subpopulation's dynamics and, for sufficiently low excitability, toward its slow limit cycle. Increasing $K$ can additionally slow down the emergent rhythms through two coupled effects: (i) stronger coupling elicits compensatory increases in ISP-driven inhibition to stabilize firing rates, and (ii) collective synchronization can "pull" the macroscopic oscillation toward a lower shared frequency even when individual nodes support faster cycles. This qualitative relationship between stronger coupling and slower collective dynamics is also consistent with phenomenological models such as Kuramoto oscillators with delays, where slower regimes can arise with higher effective inter-areal coupling [17]. Importantly, this mechanism differs from adaptive exponential neuronal models [50], where NREM slow waves arise from noise-driven transitions between up and down states governed by slow adaptation; our model does not explicitly implement bistability and instead captures sleep-associated slowing via controlled shifts in excitability and coupling within an oscillatory (limit-cycle) regime.

While the classical Jansen-Rit model can reproduce δ/θ/α rhythms, this typically occurs within restricted parameter regimes and does not capture the full-band spectral structure. Extending the model to support β/γ dynamics broadens its dynamical repertoire, opening the door to study phenomena where higher frequencies are central. For example, the model can be used to explore how attention and working memory, as well as perceptual integration/binding, are related to β/γ power modulation [51]. In addition, β/γ alterations are consistently reported in brain disorders, including neurodegenerative conditions. The ISP mechanism, combined with the multi-frequency model, allows us to simulate regimes of hypo- and hyperexcitability linked to E/I imbalance [7,52-54]. Prior studies have used PET-based amyloid and tau deposition to modulate local excitability and mimic E/I imbalance [52,26]. Our model can potentially simulate shifts from early-stage hyperexcitability and hyperconnectivity toward later hypoexcitable, spectrally slowed states, reflecting the progression from preclinical to symptomatic stages of dementia [55,56]. Further, both EEG and fMRI features can be captured, providing a multimodal modeling tool for exploring how brain excitability modulates brain function across diverse contextual and clinical conditions.

This work has several limitations. First, excitability is modulated homogeneously across all brain regions, without accounting for known spatial differences in gene expression and neurotransmitter receptor distributions that influence

local E/I balance [2,21,24,52]. A more realistic approach would involve modulating excitability in a region-specific manner based on empirical gene expression and receptor maps [3,57]. Second, several mechanisms can modulate E/I balance beyond ISP that we did not consider in this work, e.g., changes in synaptic time constants or receptor kinetics [8,28,58], as well as adaptation-driven bistability (up/down state transitions) as implemented in adaptive exponential-based neuronal models [50]. We used two rhythm-generating subpopulations in every cortical region. Although this is a parsimonious abstraction to increase spectral richness [59], it does not aim to provide detailed mechanisms, and alternative choices can be used to shape oscillatory frequencies (e.g., E/I loop delays, synaptic time constants/receptor kinetics, and axonal transmission delays). They are not explicitly modeled here and could be incorporated into future extensions. Third, the model was validated on a relatively small cohort, which may limit generalizability. Larger and more diverse datasets are needed to assess robustness, including clinical populations. Fourth, the current implementation is limited to resting-state EEG and fMRI. Extending the model to simulate task-evoked activity, e.g., task-evoked β-γ changes, would help explore context-dependent changes in connectivity and cognitive processing [19,21]. Fifth, we primarily defined canonical EEG bands by their frequency ranges and focused on spectral content and large-scale connectivity validation (EEG functional connectivity across bands and fMRI FC). Accordingly, we did not include additional fitting targets from both modalities (e.g., simultaneous EEG spectral and FC constraints together with fMRI spectral/temporal constraints), which would represent a future extension of the model. While the model can reproduce oscillations across canonical frequency bands and captures α-band power fluctuations at the whole-brain level, we did not aim to match additional defining properties of canonical rhythms such as absolute amplitudes (e.g., μV scaling), spatial topographies (e.g., occipital dominance of α), or the full temporal structure across subjects and conditions. Reproducing these features would require additional calibration steps and potentially further physiological constraints, which are beyond the scope of this study. Also, while the model reproduces state-dependent spectra and connectivity, exemplary wake/sleep time series can appear less regular than canonical α or highly synchronized δ rhythms. This is partly due to limitations in reproducing the slowest EEG components in resting-state simulations, which we did not explicitly optimize. In addition, although EEG-derived features can be fitted using only a subset of observed nodes without reducing the size of the simulated system, we adopted a reduced fitting setup for computational convenience; this simplification is not expected to qualitatively affect the present results, but a full-system fitting strategy should be considered in future work. In the sleep/wake comparison, the external input $p$ was fixed; therefore, we do not interpret ρ as the only parameter capable of reproducing these state differences, but rather as one sufficient control parameter within the parameterization explored here. For instance, in the original Jansen-Rit model [12,13], reducing the value of the input $p$ can produce slow oscillation (increasing δ and θ relative power). Finally, while the model is well-suited for testing brain stimulation and drug effects in theory, these applications remain to be validated experimentally.

Overall, we presented and characterized a biologically grounded neural mass model capable of simulating realistic EEG and fMRI dynamics through a mechanistic modulation of excitability. The model bridges basic and clinical neuroscience, offering a framework to explore brain states, disease mechanisms, and interventions.

## 4. Methods

### Ethics Statement

***ReDLat:*** Ethical approval was obtained from the institutional ethics committee at each ReDLat-participating center, and all participants provided written informed consent in accordance with the Declaration of Helsinki.

*Young connectomes:* All participants gave written informed consent before taking part, and the study protocol received approval from the SWPS University Ethical Committee in accordance with the Declaration of Helsinki.

*Sleep data:* All participants gave written informed consent. The study was approved by the local ethics committee of Goethe University Frankfurt am Main.

## 4.1 Participants and datasets

### 4.1.1 EEG resting state and structural connectivity data (ReDLat).

We used data from 45 healthy participants from the ReDLat consortium [36]. Participants' mean age was 71.2 ± 7.2 years (29 women). We considered both functional and structural connectivity data for this group. Full details can be found in our previous works [7,60], but we briefly describe data acquisition and preprocessing below.

For EEG, participants were seated in an electromagnetically shielded EEG room and instructed to stay still, awake, and with their eyes closed. As in previous works [7], resting-state EEG data were recorded for 10 minutes using a Biosemi ActiveTwo 128-channel system, with electrodes placed around the eyes to monitor blinks and movements. The signals were sampled at 1024 Hz and referenced to linked mastoids. Offline preprocessing included filtering (0.5 to 40 Hz), re-referencing to the average of all channels, and replacing malfunctioning channels using spherical interpolation [61]. Independent component analysis [62] and visual inspection were employed to correct blink artifacts and eye movements [63–66]. Localized brain areas sources were estimated using standardized low-resolution brain electromagnetic tomography (sLORETA) [67], resulting in time series data for 90 brain regions based on the AAL90 parcellation (Table 1). We retained only 82 brain areas from the AAL parcellation, including cortical regions and the hippocampus and amygdala. We excluded the remaining subcortical regions because source-reconstructed EEG is generally less reliable for deep subcortical structures within this framework, whereas hippocampal and amygdalar sources can be better estimated [68,69].

SC matrices were derived from diffusion tensor imaging (DTI) applied to diffusion-weighted imaging (DWI) data. Preprocessing was conducted using the FSL BEDPOSTX (Bayesian Estimation of Diffusion Parameters Obtained using Sampling Techniques) toolbox [70]. Post-preprocessing, each subject's data yielded a 90 x 90 matrix representing connectivity between pairs of regions defined by the AAL parcellation. The final SC matrix was obtained by averaging the individual matrices across participants. We used the same 82 brain areas as in the EEG functional data.

### 4.1.2 Structural connectivity data (connectomes of young participants).

We used SC data of healthy participants published in a previous work [71]: 31 participants from the non-video game players group, with a mean age of 24.4 ± 3.0 years, all males. The SC data acquisition involved MR imaging on a 3-Tesla MRI scanner with a 32-channel phased array head coil. DTI data were preprocessed using the Pipeline for Analyzing Brain Diffusion Images software [72]. The processing steps included artifact correction, tensor construction, and registration to the MNI space, followed by parcellation using the AAL atlas. The 90 x 90 connectivity matrices were generated by calculating the number of fibers connecting each pair of brain regions, normalized between 0 and 1, and averaged across participants.

To improve model fitting, connections were added in the SC's anti-diagonal to account for homotopic (inter-hemispheric) connections, which are typically underestimated using DTI [73] (S3 Fig). Further details on data acquisition and preprocessing can be found in [72] and [74], and further information about participants and the original studies in [71] and [42].

### 4.1.3 fMRI-EEG co-registering during wakefulness and NREM sleep.

This dataset was originally used in [40]. A total of 63 non-sleep-deprived subjects were scanned in the evening (starting from ~8:00 PM) and preliminarily included in the study. A subset of 15 individuals with fMRI volumes in all W, N1, N2, and N3 stages was included in this study. All participants were scanned in the evening and instructed to close their eyes while lying still and relaxing. 1505 volumes of T2*-weighted echo-planar images were acquired using the following parameters: TR/TE = 2,080/30 msec, matrix 64 × 64, voxel size 3 × 3 × 2 mm³, distance factor 50%, field of view [FOV] 192 mm² at 3 T (Siemens Trio). EEG recordings were simultaneously acquired with fMRI, using an EEG (modified BrainCapMR; Easycap) cap with 30 channels (sampling rate 5 kHz, low pass filter 250 Hz), along with a polysomnographic setup that included electromyography on the chin and tibia, electrocardiogram, bipolar electrooculography, and pulse oximetry. MRI artifact correction was performed using the average artifact subtraction. The fMRI data were realigned, normalized to MNI space, and spatially smoothed using Statistical Parametric Mapping (SPM8) with a Gaussian kernel (8 mm³ full width at half maximum). Sleep staging was performed by an expert according to the AASM criteria [75].

## 4.2 Functional connectivity estimation

For EEG, we filtered signals in the common EEG frequency bands: δ (0.5-4 Hz), θ (4–8 Hz), α (8–13 Hz), and β (13–30 Hz) using a 3rd-order passband Bessel filter. We used the Hilbert transform to get the signals' envelopes, and then we filtered the envelopes with a 3rd-order Butterworth high-pass filter (cut-off frequency at 0.5 Hz), and finally, we built the FC by computing the Pearson's correlation between all pairs of areas [38].

For fMRI, we first filtered the signals between 0.01-0.08 Hz using a 3rd-order Bessel filter, and then we built the FC matrix using Pearson's correlation between all pairs of areas.

## 4.3 Power spectrum analyses

For both empirical and simulated data, the EEG power spectrum was calculated using the Welch method with 2-sec length time windows with 50% overlap [76]. The power was extracted for each EEG band and divided by the total power of the spectrum between 0.5 and 30 Hz.

## 4.4 Whole-brain model

Whole-brain activity at the source level was modeled using a modified version of the Jansen-Rit neural mass model [7,12,30]. In this model, each brain region consists of two subpopulations of neural masses, each tuned to oscillate in either the α (around 10 Hz) or γ (around 45 Hz) frequency bands of the EEG spectrum. The contribution of these subpopulations to the generation of the postsynaptic potential (PSP) of pyramidal neurons is weighted by the parameter $r^a$, which reflects the proportion of α versus γ subpopulations within the brain areas, as indicated in [30]. Specifically, $r^a = 1$ indicates a full contribution from the α subpopulation, while $r^a = 0$ indicates none. Each subpopulation is composed of pyramidal neurons, along with excitatory and inhibitory interneurons.

The PSPs, $v$, are integrated and transformed into firing rates using a sigmoid function:

$$S(v) = \frac{\zeta_{max}}{1 + \exp\left(-r\left(v - v_{th}\right)\right)}$$

where $\zeta_{max}$ is the maximal firing rate output, $r$ is the slope, and $v_{th}$ is the voltage threshold. The inverse operation is conducted by a PSP block, which convolves the firing rates of the neurons with an impulse response function defined as:

$$h_E(t) = \begin{cases} Aate^{-at}, & t \geq 0 \\ 0, & t < 0, \end{cases}$$

for the excitatory PSPs, and

$$h_I(t) = \begin{cases} Bbte^{-bt}, & t \geq 0 \\ 0, & t < 0 \end{cases}$$

for inhibitory PSPs. Here, $A$ ($B$) and $a$ ($b$) correspond to the maximal amplitude and inverse characteristic time constant of the excitatory (inhibitory) PSPs, respectively. On a macroscopic level, brain areas $i$ and $j$ are connected using an SC matrix $M$ derived from DTI data. The coupling strength was scaled by a global parameter $K$ and, considering that long-range projections are primarily excitatory [77,78], connections between brain regions involved only pyramidal neurons [7]. Each region received background input $p(t)$, with values randomly sampled from a normal distribution with a mean $\langle p(t) \rangle = 220$ Hz and a standard deviation $\sigma_p = 31$ Hz. The complete system of equations for the α subpopulation was:

$$\frac{dx_{0,i}^{\alpha}(t)}{dt} = y_{0,i}^{\alpha}(t)$$

$$\frac{dy_{0,i}^{\alpha}(t)}{dt} = A^{\alpha}a^{\alpha}S\left(x_{1,i}(t) - x_{2,i}(t)\right) - 2a^{\alpha}y_{0,i}^{\alpha}(t) - a^{\alpha^2}x_{0,i}^{\alpha}(t)$$

$$\frac{dx_{1,i}^{\alpha}(t)}{dt} = y_{1,i}^{\alpha}(t)$$

$$\frac{dy_{1,i}^{\alpha}(t)}{dt} = A^{\alpha}a^{\alpha}\left(p_i(t) + C_2 S\left(C_1 x_{0,i}(t)\right)\right) + KC\sum_{j=1, j\neq i}^{N} M_{ij} S\left(x_{1,j}(t) - x_{2,j}(t)\right) - 2a^{\alpha}y_{1,i}^{\alpha}(t) - a^{\alpha^2}x_{1,i}^{\alpha}(t)$$

$$\frac{dx_{2,i}^{\alpha}(t)}{dt} = y_{2,i}^{\alpha}(t)$$

$$\frac{dy_{2,i}^{\alpha}(t)}{dt} = B^{\alpha}b^{\alpha}\left(C_4 S\left(C_3 x_{0,i}(t)\right)\right) - 2b^{\alpha}y_{2,i}^{\alpha}(t) - b^{\alpha^2}x_{2,i}^{\alpha}(t)$$

The first pair of equations model the excitatory feedback loop, the second pair models the outputs of pyramidal neurons, and the third represents the inhibitory feedback loop. Neuron populations were connected via constants $C_1$, $C_2$, $C_3$ and $C_4$, which are scaled by a local connectivity constant, $C = 135$, with $C_1 = C$, $C_2 = 0.8C$, $C_3 = 0.25C$, and $C_4 = 0.25C$. Identical equations apply to the $\gamma$ subpopulations, differing only by the $\gamma$ superscript. The final output of the model produced EEG-like signals in source space, featuring a richer power spectrum. The model's parameters are summarized in Table 2.

The rationale of combining these two subpopulations is grounded in the organization of cortical layers, where different neuronal subpopulations exhibit distinct input-output dynamics and generate frequency-specific rhythms [79]. Superficial layers tend to produce faster oscillations, such as $\beta$ and $\gamma$, while deeper and granular layers are associated with slower $\alpha$ and $\theta$ activity [80]. Here, we use two subpopulations as a parsimonious modeling choice to broaden the nodal spectral repertoire while keeping the whole-brain model tractable for fitting. This does not mean that this approach should be treated as an account of the unique origin of each rhythm in every region. Similar multi-timescale approaches have been explored previously in whole-brain modeling to increase spectral richness at each node [59]. In our model, the EEG reflects a weighted summation of these laminar contributions, and the resulting broadband spectrum emerges from the combined activity of spatially and synaptically distinct rhythm-generating circuits. The state variables $x_0(t)$, $x_1(t)$, and $x_2(t)$ are defined as a weighted contribution of $\alpha$ and $\gamma$ subpopulations:

$$x_0(t) = r^{\alpha}x_0^{\alpha}(t) - (1 - r^{\alpha})x_0^{\gamma}(t)$$

$$x_1(t) = r^{\alpha}x_1^{\alpha}(t) - (1 - r^{\alpha})x_1^{\gamma}(t)$$

$$x_2(t) = r^{\alpha}x_2^{\alpha}(t) - (1 - r^{\alpha})x_2^{\gamma}(t)$$

**Table 2. Model's parameters.**

| Parameter | Values | Units | Meaning |
|---|---|---|---|
| $A^{\alpha}$ | 3.9 | mV | Maximal amplitude of EPSPs ($\alpha$ population) |
| $A^{\gamma}$ | 32.5 $a^{\gamma}$/1000 | mV | Maximal amplitude of EPSPs ($\gamma$ population) |
| $B^{\alpha}$ | 26.4 | mV | Maximal amplitude of IPSPs ($\alpha$ population) |
| $B^{\gamma}$ | 440 $b^{\gamma}$/1000 | mV | Maximal amplitude of IPSPs ($\gamma$ population) |
| $a^{\alpha}$ | 120 | sec$^{-1}$ | Inverse characteristic time constant for EPSPs ($\alpha$ population) |
| $a^{\gamma}$ | 660 | sec$^{-1}$ | Inverse characteristic time constant for EPSPs ($\gamma$ population) |
| $b^{\alpha}$ | 60 | sec$^{-1}$ | Inverse characteristic time constant for IPSPs ($\alpha$ population) |
| $b^{\gamma}$ | 330 | sec$^{-1}$ | Inverse characteristic time constant for IPSPs ($\gamma$ population) |
| $r^{\alpha}$ | 0.5, [0,1] | Unitless | Proportion of α neurons |
| $\xi_{max}$ | 5 | sec$^{-1}$ | Maximum firing rate of the sigmoid function |
| $r$ | 0.56 | mV$^{-1}$ | Slope of sigmoid function |
| $v_{th}$ | 6 | mV | Threshold of the sigmoid function |
| $M$ | [0,1] | Unitless | Structural connectivity matrix |
| $C$ | 135 | Unitless | Local connectivity |
| $C_1$ | C | Unitless | Excitatory-to-pyramidal neurons connectivity |
| $C_2$ | 0.8C | Unitless | Pyramidal-to-excitatory neurons connectivity |
| $C_3$ | 0.25C | Unitless | Inhibitory-to-pyramidal neurons connectivity |
| $C_4$ | 0.25C, variable | Unitless | Pyramidal-to-inhibitory neurons connectivity |
| $\langle p(t) \rangle$ | 220 | sec$^{-1}$ | External input mean |
| $\sigma_p$ | 31 | sec$^{-1}$ | External input standard deviation |
| $\rho$ | 2.5, [2.0,3.0] | sec$^{-1}$ | Target firing rate for plasticity |
| $\tau$ | 2, [0.01, 50] | sec | Inverse of plasticity's learning rate |
| $\beta$ | 1 | Unitless | Bounding exponent of plasticity |
| $K$ | [0,3] | Unitless | Global coupling |
| $\Delta t$ | 1 | msec | Integration step |
| $t_{eq}$ | 10 | sec | Discarded simulation time |
| $t_{sim}$ | [180, 660] | sec | Total simulation time |
| $t_{max}$ | [120, 600] | sec | Final time length of simulations |

The model EEG-like signal can be computed as [7,30]:

$$EEG(t) = x_1(t) - x_2(t)$$

Additionally, following [9], we integrated inhibitory synaptic plasticity into the model. This mechanism controls the firing rate of pyramidal neurons, preventing hyperexcitability, which can occur when global coupling increases. Synaptic plasticity was introduced as an additional differential equation:

$$\tau \frac{dC_4(t)}{dt} = \zeta_{inh}(t) \left( \zeta_{pyr}(t) - \rho \right) \left( C_4(t)/C - C_{4,min}/C \right)^{\beta}$$

This equation dynamically updates feedback inhibition to regulate the firing rate of pyramidal neurons, thus preventing saturation of the sigmoid function (hyperexcitability). Here, $\tau$ represents the inverse learning rate, $\zeta_{inh}(t)$ and $\zeta_{pyr}(t)$ are the firing rates of inhibitory interneurons and pyramidal neurons at time $t$, respectively, $\rho$ is the target firing rate, and $\beta$ is a

bounding exponent controlling convergence to $C_{4,min}$. We used $\beta = 1$ (soft bound), though other values are possible [81]. The key parameters for plasticity are the learning rate, $\tau$, and the target firing rate, $\rho$, which were set by default to $\tau = 2\,\mathrm{sec}$ and $\rho = 2.5$ Hz [9,35]. The firing rates are defined as

$$\zeta_{inh}(t) = S\left(C_3 x_0(t)\right)$$

$$\zeta_{exc}(t) = S\left(C_1 x_0(t)\right)$$

$$\zeta_{pyr}(t) = S\left(x_1(t) - x_2(t)\right)$$

We solved the system of differential equations using the Euler-Maruyama method with a 1 msec integration step. The filtering and FC estimation procedures were the same as those used for the empirical signals.

## 4.5  Hemodynamic model

We simulated fMRI-like signals using the pyramidal firing rates $\zeta_{pyr,i}(t)$ and a generalized hemodynamic model as presented by Stephan et al. [43]. In this model, an increase in the firing rate $\zeta_{pyr,i}(t)$ initiates a vasodilatory response $s_i$ leading to blood inflow $f_i$, and subsequent changes in blood volume $v_i$ and deoxyhemoglobin content $q_i$. The system of differential equations governing this process is

$$\frac{ds_i(t)}{dt} = \zeta_{pyr,i}(t) - \frac{s_i(t)}{\tau_s} - \frac{f_i(t) - 1}{\tau_f}$$

$$\frac{df_i(t)}{dt} = s_i(t)$$

$$\frac{dv_i(t)}{dt} = \frac{f_i(t) - v_i(t)^{\frac{1}{\kappa}}}{\tau_v}$$

$$\frac{dq_i(t)}{dt} = \frac{\frac{f_i(t)\left(1 - (1-E_0)^{\frac{1}{f_i(t)}}\right)}{E_0} - \frac{q_i(t)v_i(t)^{\frac{1}{\kappa}}}{v_i(t)}}{\tau_q}$$

where the time constants $\tau_s = 0.65$, $\tau_f = 0.41$, $\tau_v = 0.98$, and $\tau_q = 0.98$ correspond to signal decay, blood inflow, blood volume, and deoxyhemoglobin content, respectively. The stiffness constant $\kappa$ (resistance of the veins to blood flow) and the resting-state oxygen extraction rate $E_0$ were set to $\kappa = 0.32$ and $E_0 = 0.4$. The BOLD-like signal for the node $i$, denoted as $B_i(t)$, is a nonlinear function of $q_i(t)$ and $v_i(t)$:

$$B_i(t) = V_0 \left[ k_1 \left(1 - q_i(t)\right) + k_2 \left(1 - \frac{q_i(t)}{v_i(t)}\right) + k_3 \left(1 - v_i(t)\right) \right]$$

where $V_0 = 0.04$ represents the fraction of deoxygenated venous blood at rest, and $k_1 = 2.77$, $k_2 = 0.2$, $k_3 = 0.5$ are kinetic constants.

We solved the system of differential equations using the Euler method with a 10-msec integration step. The resulting fMRI-like signals were then down-sampled to match the empirical signal's sampling rate (TR = 2.08 sec). The filtering and FC estimation followed the same procedures applied to the empirical signals.

## 4.6 Simulations

We first simulated the activity of an isolated node of the model ($K = 0$) without plasticity. There, we simulated 180 sec and discarded the first 60 sec (for a final length of 120 sec), downsampled the signals to a 200 Hz sampling rate (to reduce memory consumption), and calculated the Normalized Power Spectrum (NPS, obtained using the Welch method and dividing by the maximum value of spectral density). Simulations were run against variations of $r^a$ in 41 equidistant values between 0 and 1 (Fig 3).

We then ran a whole-brain simulation, where nodes represent brain areas that are connected using the ReDLat SC matrix. We simulated 660 sec, discarded the first 60 sec, and obtained the NPS (averaged across brain areas) against variations in different parameters (average of 50 seeds in Fig 4): we varied $K$ over 41 equidistant values between 0 and 1 (upper right, fixing $r^a = 0.5$, $\rho = 2.5$ Hz, $\tau = 2$ sec); $\rho$ over 41 equidistant values between 1 and 4 Hz (lower left, fixing $r^a = 0.5$, $K = 0.5$, $\tau = 2$ sec); and $\tau$ over 41 equidistant and log spaced values between 0.01 and 50 sec (lower right, fixing $r^a = 0.5$, $\rho = 2.5$ Hz, $K = 0.5$).

Using the same whole-brain simulations described above (ISP enabled), we quantified how the spatial distribution of feedback inhibition relates to the structural connectivity topology. For each simulation, we computed the nodal strength of each brain area as the sum of its structural connectivity weights (row-sum of the SC matrix). We then computed the time-averaged feedback inhibition (mean $C_4$) for each node by averaging $C_4$ over time after 20 sec simulations, discarding the initial 5 sec transient, using 10 random seeds. For each value of the global coupling $K$, we fitted a linear regression between nodal strength and mean $C_4$ across nodes and stored the regression slope (Fig 4). Unless otherwise stated, these analyses were performed with $r^a = 0.5$, $\rho = 2.5$ Hz, and $\tau = 2$ sec.

To quantify how the ISP time constant $\tau$ affects adaptation dynamics, we ran noise-free whole-brain simulations ($\sigma = 0$) with ISP enabled and fixed parameters $K = 0.5$, $r^a = 0.5$, and $\rho = 2.5$ Hz, using only one random seed (noise-free simulation). Simulations had a total duration of 120 sec. We swept $\tau$ over the values $\tau \in [10^{-3}, 10^{-2}, 10^{-1}, 1, 10, 100]$ sec. For each $\tau$, we computed the ROI-averaged $C_4(t)$ and defined a convergence timescale as the first time at which $C_4(t)$ reached 63.2% ($1 - 1/e$) of its total change from an initial baseline $C_4^0$ to its final steady value $C_4^\infty$, where $C_4^0$ was estimated from the first 2 sec and $C_4^\infty$ from the last 10 s of the simulation (Fig 4).

We fitted the model to the average FC matrices of healthy individuals in the ReDLat dataset [60], simulating 660 sec and discarding the first 60 sec of adaptation. We fixed $\rho = 2.5$ Hz and $\tau = 2$ sec and simultaneously varied the model's global coupling $K$ and $r^a$ between 0 and 1 (41 values each), first without ISP and then with ISP (Fig 5B). This was carried out by fitting the source-EEG envelope FC, after filtering the signal in the different EEG frequency bands, and downsampling to a 100 Hz sampling rate.

For the sleep EEG-fMRI dataset (Fig 8) [82], we swept the coupling parameter, $K$, from 0 to 3 (41 values in total), and the target value for ISP, $\rho$, from 2 to 3 (41 values as well), fixing $r^a = 0.5$ and $\tau = 2$ sec. We simulated 660 seconds and discarded the first 60 sec for stabilization. We used the pyramidal neurons' firing rates to simulate fMRI-like signals through the Balloon-Windkessel model [43], averaging FCs across the 50 simulation seeds. We used firing rates for both α and γ subpopulations scaled by $r^a$. The SC matrix came from the connectome of the young participants dataset [11].

## 4.7 Model fitting

For EEG and fMRI FC matrices, we maximized the SSIM [83], used to compare empirical and simulated FC matrices in whole-brain simulations [73].

For the simultaneous EEG and fMRI dataset, after running the model, we aimed to fit the EEG power spectrum and fMRI FC. First, we calculated the relative power of each EEG band using the Welch method, generating a spectral vector of three entries (relative θ, α, and β power) for empirical and simulated data. Then, we compared the distance between the empirical and simulated relative power using the Clarkson Distance (say, x is the simulated power vector and y the empirical power vector):

$$\lambda(x, y) = \frac{1}{\sqrt{2}} \left| \frac{x}{|x|} - \frac{y}{|y|} \right|$$

where $|x|$ is the Euclidean norm of x. We took $1 - \lambda(x, y)$ as the goodness of fit between spectral vectors $x, y$.

Finally, we calculated the distance between empirical and simulated spectral vectors for each parameter combination and further analyzed only parameter combinations with a goodness of fit $\geq 0.85$. From this subset of parameters, we maximized the SSIM between empirical and simulated FCs.

## 4.8 Bifurcation analysis

Bifurcation analysis was performed using the XPP-Auto software, and the figures were made using custom code in Python.

## 4.9 Statistical analysis

Given that statistical p-values could be artificially inflated by sample size in computer simulations (varying the number of seeds), instead of statistical tests, we report results using Cohen's D for effect size. Usually, Cohen's D is interpreted as very small ($|D| < 0.2$), small ($0.2 < |D| < 0.5$), moderate ($0.5 < |D| < 0.8$), large ($0.8 < |D| < 1.2$), very large ($1.2 < |D| < 2$) [84].

## Supporting information

**S1 Fig. Extended bifurcation analysis across homeostatic targets.** For the Jansen-Rit model with homeostatic plasticity, the bifurcation diagram (left), associated main oscillation frequency (center), and two representative activity traces with their corresponding power spectral density (PSD; right) are shown for multiple pyramidal firing-rate targets (rows: $\rho = 2.5, 2.9, 3.0, 3.2, 3.3$ Hz; indicated in each left panel). In the bifurcation diagrams, thick red and black lines denote stable and unstable fixed points, respectively, while green and blue curves indicate maxima and minima of stable (green) and unstable (blue) periodic orbits. The same color convention is used in the center panels to report the oscillation frequency (Hz) along the corresponding periodic branches. In the right panels we show sample time series and PSD, at the labeled input values $p$. Dots denote bifurcations: black = Hopf bifurcation; purple = saddle-node (either fixed points or limit cycles); blue = Torus or Neimar-Sacker bifurcation; red = branching or pitchfork bifurcation.
(PDF)

**S2 Fig. Bifurcation analysis with respect to the target firing rate $\rho$ at fixed input $p$. For the Jansen-Rit model with homeostatic plasticity, bifurcation diagrams are shown as a function of the target $\rho$ for three representative constant input values (top to bottom: $p = 100, 200, 240$).** Left panels show the equilibrium variable $x_0$ (excitatory input to pyramidal neurons) and the extrema of periodic orbits as $\rho$ is varied: thick red and black lines indicate stable and unstable fixed points, respectively, and green/blue curves indicate maxima and minima of stable (green) and unstable (blue) periodic orbits. In the right panels we report the main oscillation frequency (Hz) along the corresponding periodic branches using the same color convention. Dots denote bifurcations: black = Hopf; purple = saddle-node (of fixed points or limit cycles); blue = torus/Neimark-Sacker; red = branching/pitchfork bifurcation.
(PDF)

**S3 Fig. Structural connectivity (SC) matrices used in whole-brain simulations. A)** The original SC matrix (young connectome). **B)** Modified SC matrix after reinforcing homotopic inter-hemispheric connections along the anti-diagonal. **C)** Difference matrix (optimized minus original).
(PDF)

## Author contributions

**Conceptualization:** Carlos Coronel-Oliveros, Patricio Orio, Agustín Ibáñez.

**Data curation:** Natalia Kowalczyk-Grębska, Raul Gonzalez-Gomez.

**Formal analysis:** Carlos Coronel-Oliveros, Fernando Lehue, Patricio Orio.

**Funding acquisition:** Agustín Ibáñez.

**Investigation:** Carlos Coronel-Oliveros, Patricio Orio, Agustín Ibáñez.

**Methodology:** Carlos Coronel-Oliveros, Fernando Lehue, Patricio Orio, Agustín Ibáñez.

**Project administration:** Patricio Orio, Agustín Ibáñez.

**Resources:** Patricio Orio, Agustín Ibáñez.

**Software:** Carlos Coronel-Oliveros.

**Supervision:** Patricio Orio, Agustín Ibáñez.

**Validation:** Carlos Coronel-Oliveros, Fernando Lehue, Patricio Orio, Agustín Ibáñez.

**Visualization:** Carlos Coronel-Oliveros, Fernando Lehue, Patricio Orio.

**Writing – original draft:** Carlos Coronel-Oliveros, Fernando Lehue, Rubén Herzog, Iván Mindlin, Marilyn Gatica, Natalia Kowalczyk-Grębska, Vicente Medel, Josephine Cruzat, Raul Gonzalez-Gomez, Hernán Hernandez, Enzo Tagliazucchi, Pavel Prado, Patricio Orio, Agustín Ibáñez.

**Writing – review & editing:** Carlos Coronel-Oliveros, Fernando Lehue, Rubén Herzog, Iván Mindlin, Marilyn Gatica, Natalia Kowalczyk-Grębska, Vicente Medel, Josephine Cruzat, Raul Gonzalez-Gomez, Hernán Hernandez, Enzo Tagliazucchi, Pavel Prado, Patricio Orio, Agustín Ibáñez.

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
