## [Decision Letter · Decision Letter 0]

19 Nov 2025

PCOMPBIOL-D-25-01699

A multi-frequency whole-brain neural mass model with homeostatic feedback inhibition

PLOS Computational Biology

Dear Dr. Ibáñez,

Thank you for submitting your manuscript to PLOS Computational Biology. After careful consideration, we feel that it has merit but does not fully meet PLOS Computational Biology's publication criteria as it currently stands. Therefore, we invite you to submit a revised version of the manuscript that addresses the points raised during the review process.

We look forward to receiving your revised manuscript.

Kind regards,

Sacha Jennifer van Albada

Academic Editor

PLOS Computational Biology

Hugues Berry

Section Editor

PLOS Computational Biology

**Journal Requirements:**

At this stage, the following Authors/Authors require contributions: Carlos Coronel-Oliveros, Josephine Cruzat, Marilyn Gatica, Raul Gonzalez-Gomez, Hernán Hernandez, Rubén Herzog, Natalia Kowalczyk-Grębska, Fernando Lehue, Vicente Medel, Iván Mindlin, Patricio Orio, Pavel Prado, Enzo Tagliazucchi, and Agustín Ibáñez. Please ensure that the full contributions of each author are acknowledged in the "Add/Edit/Remove Authors" section of our submission form.

2) Some material included in your submission may be copyrighted. According to PLOSu2019s copyright policy, authors who use figures or other material (e.g., graphics, clipart, maps) from another author or copyright holder must demonstrate or obtain permission to publish this material under the Creative Commons Attribution 4.0 International (CC BY 4.0) License used by PLOS journals. Please closely review the details of PLOSu2019s copyright requirements here: PLOS Licenses and Copyright. If you need to request permissions from a copyright holder, you may use PLOS's Copyright Content Permission form.

Potential Copyright Issues:

i) Figure 1A. Please confirm whether you drew the images / clip-art within the figure panels by hand. If you did not draw the images, please provide (a) a link to the source of the images or icons and their license / terms of use; or (b) written permission from the copyright holder to publish the images or icons under our CC BY 4.0 license. Alternatively, you may replace the images with open source alternatives. See these open source resources you may use to replace images / clip-art:

ii) Figure 1B appears to have been modified from a previously published figure. Please provide written permission from the copyright holder to publish this under our CC-BY 4.0 license, or remove the figure / replace the image. Please note we do not recommend using standard request forms available on Publishers' websites, as they grant single use rather than republication under an open access license.

3) In the online submission form, you indicated that  Raw data are available on request from Drs. Agustín Ibáñez (agustin.ibanez@gbhi.org) and Natalia Kowalczyk-Grębska (nkowalczyk@swps.edu.pl). . All PLOS journals now require all data underlying the findings described in their manuscript to be freely available to other researchers, either

1. In a public repository

2. Within the manuscript itself

3. Uploaded as supplementary information.

4) Please amend your detailed Financial Disclosure statement. This is published with the article. It must therefore be completed in full sentences and contain the exact wording you wish to be published.

5) Your current Financial Disclosure states, "Yes ↳ Please add funding details. AI is supported by grants from the Multi-partner consortium to expand dementia research in Latin America [ReDLat, supported by Fogarty International Center (FIC), National Institutes of Health, National Institutes of Aging (R01s AG075775, AG057234, AG082056 and AG083799, CARDS-NIH 75N95022C00031), Alzheimer's Association (SG-20-725707), Rainwater Charitable Foundation – The Bluefield project to cure FTD, and Global Brain Health Institute)], ANID/FONDECYT Regular (1250091, 1210195, 1210176, and 1220995); ANID/PIA/ANILLOS ACT210096; FONDEF ID20I10152, and ANID/FONDAP 15150012. JC is supported by ANID (FONDECYT Postdoctorado #3240042; FONDECYT de Exploración #13240170). This research was supported by the National Science Centre (Poland) Grant: 2013/11/N/HS6/01335, in the years 2013–2017 (to NK-G). The contents of this publication are solely the authors' responsibility and do not represent the official views of these institutions. RH was partially supported by the Ramón y Cajal Fellowship (RYC2022-035106-I) from FSE/Agencia Estatal de Investigación (AEI), Spanish Ministry of Science and Innovation, and the María de Maeztu Program for units of Excellence in R&D, grant CEX2021-001164-M/10.13039/501100011033. ↳ Please select the country of your main research funder (please select carefully as in some cases this is used in fee calculation). UNITED STATES - US".

However, your funding information on the submission form indicates receiving no funds.

Please indicate by return email the full and correct funding information for your study and confirm the order in which funding contributions should appear. Please be sure to indicate whether the funders played any role in the study design, data collection and analysis, decision to publish, or preparation of the manuscript.

6) Kindly revise your competing statement in the online submission form to align with the journal's style guidelines: 'The authors declare that there are no competing interests.'

**Reviewers' comments:**

Reviewer's Responses to Questions

**Comments to the Authors:**

Reviewer #1: The authors propose and evaluate a modification of the Jansen Rit neural mass model combining two extensions : incorporation of two subpopulations at each node providing oscillatory activity on two different frequencies, and a plasticity mechanism to adjust the inhibitory freedback to maintain target firing rates.

In particular, the work presented here builds on previously published models implementing the two individual extensions, evaluates the impact of the plasticity on the bifurcation structure, and demonstrates the capacity of the resulting model to reproduce functional connectivity for both EEG and fMRI in awake and sleep brain states. The manuscript is overall easy to follow and the presented analysis is clear.

If I understand it correctly, both of the extensions have been published before: multiple populations embodying different kinetics in e.g. (1, 2), and the plasticity mechanism introduced in (3). Currently, it is difficult to appreciate what contributions the authors make here before engaging with the manuscript in depth. More precise formulation of the contributions already in the abstract and introduction would help the reader to understand these relationships.

The dependence of the bifurcation structure on the target firing rate presented in Figure 2BC is an interesting feature of the model. It would be great to provide the diagrams for more values of target firing rate (e.g. as supplementary) to help to build the intuition on the sensitivity wrt to this parameter.

In addition, the time-scale of plasticity mechanism doesn't seem to play a role (Fig. 4C), which I'd think means that after the initial transient the $C_4$ parameter remains constant. Is this the case? If so, isn't then the plasticity mechanism effectively part of the fitting procedure? In other words, wouldn't it be possible to reach similar results to adjust $C_4$ together with changing coupling strength?

The discussion of the reproduction of sleep and wake activity is not sufficient in my opinion. The authors state, that their "approach aligns with previous computational work modeling sleep-like dynamics through chemical neuromodulation", however I don't see this alignment especially with respect to the work utilizing the AdEx+adaptation model (refs 52 and 55). There the sleep-like slow-waves are arising from up- and down-state transitions driven by the fluctuation of the adaptation on the slow time-scale, whereas in this work, the sleep-like dynamics are result of the dominance of the slower oscillating population. As the bistable approach reflects the empirical observations better, a more precise discussion would be helpful for the readers to be able to appreciate the differences.

Some of the methodological choices are stated, but not justified. It is not clear to me, why only 82 regions of the AAL parcellation were used in the EEG simulations, but all 90 regions were used for fMRI simulations. Also, the homotopic connections were strenghtened to compensate underestimation by DTI, however more details would be helpful to understand this step (e.g. a plot of the mask added to the SC).

And lastly, I'm not sure if the claims of the model suitability for a broad range of applications are fully supported (chemical neuromodulation, drug delivery, altered states of consciousness; bridges basic and clinical neuroscience). I believe a more grounded framing of the work would not take away from its appeal to the target audience.

In addition, I would have the following specific comments:

- section 2.2 refers to node-level PSP (abbreviation defined later), but it is not clear how that is defined in the multifrequency model (weighted sum of )

- the $x_{1,i}$ and $x_{2,i}$ PSPs are not defined and from the text it is not clear how they are computed exactly.. Similarly, $r^\alpha$ is not appearing in the provided equations.

- in section 2.4, the typesetting of 1-Clarkson's distatance is confusing

- there is a typo at the end of the first paragraph of section 4.2

- local connectivity constant C is mentioned in the text, but present in the equations

(1) Sanchez-Todo, Roser, et al. "A physical neural mass model framework for the analysis of oscillatory generators from laminar electrophysiological recordings." _NeuroImage_ 270 (2023): 119938.

(2) David, Olivier, and Karl J. Friston. "A neural mass model for MEG/EEG:: coupling and neuronal dynamics." _NeuroImage_ 20.3 (2003): 1743-1755.

(3) Coronel-Oliveros, Carlos, et al. "Viscous dynamics associated with hypoexcitation and structural disintegration in neurodegeneration via generative whole-brain modeling." _Alzheimer's & Dementia_ 20.5 (2024): 3228-3250.

Reviewer #2: Attachment

Reviewer #3: Summary:

The authors present a novel extension of the classical Jansen–Rit neural mass model, aiming to increase its flexibility and biological plausibility.

In contrast to the standard Jansen–Rit formulation, each neural mass is subdivided into two interconnected subpopulations, expressing fast and slow rhythmic activity, which allows it to switch between them

and provide a broader spectrum of frequencies. Furthermore, the authors implement a feedback inhibition control mechanism to adjust inhibitory gain in order to maintain the system near desired dynamical regimes.

The model is first analyzed at the single-node level to characterize its intrinsic dynamical repertoire and bifurcation structure. Subsequently, it is embedded into a whole-brain network using source-level functional connectivity derived from EEG recordings of healthy individuals. The authors further apply the model to multimodal EEG–fMRI data acquired during sleep, exploring whether the modified dynamics can capture state-dependent changes in brain activity.

Conclusion:

The authors combine two recent extensions of the Jansen–Rit model into a unified framework that offers a flexible and potentially valuable tool for whole-brain modeling. The approach is novel and likely to be helpful for the community, as it improves the dynamical richness and stability of large-scale simulations. The manuscript thoroughly examines the model and provides rigorous validation across various scales and datasets. Meanwhile, the manuscript would benefit from further exploration of the suggested model and a clearer justification of several modeling choices to strengthen confidence in the proposed framework and to provide a more detailed understanding of its advantages and limitations.

Questions to the Authors:

1. In the Introduction (p. 9), you refer to “hyperexcitability” as a known problem. Could

you explain this in more detail? This phenomenon depends strongly on the specific model

formulation and fitting strategy, and different approaches exist to address overshooting

excitation (as you also note later in the Discussion). Additionally, what is the conceptual

link between increased excitation, plasticity, and the feedback-inhibition mechanism

introduced later?

2. You state that the two subpopulations are tuned to oscillate on fast limit cycles in either

the alpha or gamma range. What are the corresponding slow-limit-cycle frequencies for

both subpopulations? In the classical Jansen–Rit model, the slow cycle typically lies in

the theta/delta range—how is this preserved or modified for the population tuned to

gamma?

3. The use of two subpopulations per cortical column is a pragmatic strategy to broaden the

frequency repertoire; however, EEG frequency-band origins remain an active area of

debate and exhibit regional variability. Could you provide a more detailed justification

for reducing the origin of different brain rhythms to the same two subpopulations in each

cortical column and discuss the implications of this simplification?

4. Please clarify which empirical or theoretical phenomena specifically require extending

the model to higher frequencies. The classical Jansen–Rit model can already produce

alpha, theta, and delta rhythms (although just within a restricted parameter range), which

are also the primary rhythms relevant for, e.g., the sleep experiment included in this

study. Therefore, a brief review of why beta and gamma frequencies are important, such

as in plasticity and learning, would provide helpful context.

5. Regarding Results 2.2: Why are only the amygdala and hippocampus included among

subcortical regions?

6. Also in 2.2: While it is intuitive that high ! leads to slower oscillations, why do high

values of also produce slower dynamics? A brief explanation referencing the

bifurcation structure described in Section 2.1 might help to understand this behavior

better.

7. In Section 2.3, canonical EEG bands appear to be characterized solely by their frequency

ranges. However, in fact, they are also defined by characteristic amplitudes, spatial

topographies, and temporal dynamics (e.g., alpha amplitude fluctuations, occipital

dominance, and a typical amplitude of 20–60 µV). Since the original Jansen–Rit model

captures some of these aspects, to what extent does your modified model reproduce these

additional features?

8. In 2.3, why do you consider only delta, theta, alpha, and beta bands for EEG-derived FC,

even though one of the motivations for the multi-frequency model is to include gamma

oscillations?

9. Please justify the choice of SSIM as the similarity metric. Why was it preferred over

other metrics, and could you also provide correlation coefficients to offer readers a

clearer sense of similarity magnitude?

10. Why do the analyses in 2.3 rely exclusively on FC metrics? Could additional metrics,

such as FCD or frequency spectra, be incorporated, mainly since spectral fitting is used

later in the study?

11. Relatedly, you state that the system is “unable to reach” certain areas of parameter space

without feedback inhibition. Would these areas become accessible if an additional fitting

target were included—such as a synchrony-related metric that counteracts the bias toward

hypersynchronized regimes, usually mainly driven by the genuine similarity between

functional and structural connectivity? In Section 2.4, you note to yourselves that

additional targets (e.g., a spectral constraint) can "mask" parts of the parameter space.

12. Figure 6: Panels C and D appear without labels; please add the corresponding letters to

match the caption.

13. In Section 2.4, why are the fitting targets limited to EEG frequency and fMRI-derived

FC? Why not use spectral and FC targets from both EEG and fMRI?

14. For Figure 8, please provide the correlations between empirical and simulated FCs, and

comment on the degree of inter-individual variability.

15. Figure 8C: The (exemplary) simulated wake signal does (from first view) seem not to

resemble typical alpha activity (as it is too irregular), and the deep-sleep signal also lacks

the highly regular, synchronized delta waves observed empirically. How is this in the

empirical dataset used here? Please comment on the model's ability to capture these

qualitative features and provide the corresponding empirical examples of time series for

comparison.

16. Figure 8D: In NREM stage 3, please include the empirical spectra for comparison, as

empirical power spectra typically show narrow-band, high-amplitude delta dominance

17. On page 14, “Janse-Rit” should be corrected to “Jansen-Rit.”

18. On page 14, you state: "Superficial layers tend to produce faster oscillations such as α

and θ, while deeper and granular layers are associated with slower β and γ activity." In

how far are alpha and theta faster than beta and gamma?

19. Page 14 again states that the model can express frequencies from delta to gamma. Why,

then, are gamma-band results not included in Sections 2.3 and 2.4?

**Have the authors made all data and (if applicable) computational code underlying the findings in their manuscript fully available?**

Reviewer #1: Yes

Reviewer #2: Yes

Reviewer #3: Yes

PLOS authors have the option to publish the peer review history of their article (what does this mean?). If published, this will include your full peer review and any attached files.

Reviewer #1: **Yes:**Jan Fousek

Reviewer #2: **Yes:**Damien Depannemaecker

Reviewer #3: **Yes:**Leon Stefanovski

**Figure resubmission:**

After uploading your figures to PLOS’s NAAS tool - https://ngplosjournals.pagemajik.ai/artanalysis, NAAS will process the files provided and display the results in the "Uploaded Files" section of the page as the processing is complete. If the uploaded figures meet our requirements (or NAAS is able to fix the files to meet our requirements), the figure will be marked as "fixed" above. If NAAS is unable to fix the files, a red "failed" label will appear above. When NAAS has confirmed that the figure files meet our requirements, please download the file via the download option, and include these NAAS processed figure files when submitting your revised manuscript. **Reproducibility:**

---

## [Decision Letter · Decision Letter 1]

21 Apr 2026

PCOMPBIOL-D-25-01699R1

A multi-frequency whole-brain neural mass model with homeostatic feedback inhibition

PLOS Computational Biology

Dear Dr. Ibáñez,

Thank you for submitting your manuscript to PLOS Computational Biology. While the reviewers were happy with your revision, they identified a few minor remaining points to be addressed. Therefore, we invite you to submit a revised version of the manuscript that addresses the points raised during the review process.

We look forward to receiving your revised manuscript.

Kind regards,

Sacha Jennifer van Albada

Academic Editor

PLOS Computational Biology

Hugues Berry

Section Editor

PLOS Computational Biology

**Journal Requirements:**

At this stage, the following Authors/Authors require contributions: Carlos Coronel-Oliveros, Josephine Cruzat, Marilyn Gatica, Raul Gonzalez-Gomez, Hernán Hernandez, Rubén Herzog, Natalia Kowalczyk-Grębska, Fernando Lehue, Vicente Medel, Iván Mindlin, Patricio Orio, Pavel Prado, Enzo Tagliazucchi, and Agustín Ibáñez. Please ensure that the full contributions of each author are acknowledged in the "Add/Edit/Remove Authors" section of our submission form.

**Reviewers' comments:**

Reviewer's Responses to Questions

**Comments to the Authors:**

Reviewer #1: I'd like to thank the authors for the revision, my comments were appropriately addressed.

I have one minor comment regarding the fitting of EEG source data. There is no need to reduce the size of the simulated system for fitting EEG data features derived from a subset of the nodes. However I believe the results would not change qualitatively in present work, the authors may take this into consideration either in the discussion (if additional revisions are made), or in future work..

Reviewer #2: In this resubmitted version, the authors addressed most of the concerns. A few points remain to be improve:

- The authors deflected this rather than answered it directly. The letter politely but clearly asks for one or two sentences clarifying whether the prior multi-frequency model (without ISP) also suffered from hyperexcitability, a simple but scientifically important clarification.

- The overclaim was removed, but the sensitivity question was never addressed. A one-sentence acknowledgment that p was held fixed in the sleep/wake comparison would be useful.

- Few typos and missing information:

Intro: “tunning”

in 2.4 Dual fitting to EEG and fMRI -> “(define grid)” not defined….”

Etc…

Reviewer #3: Thank you very much for this detailed and comprehensive revision. All my points have been addressed adequately, and in my view the impact and quality of the manuscript has further improved a lot. I have no further questions to add.

**Have the authors made all data and (if applicable) computational code underlying the findings in their manuscript fully available?**

Reviewer #1: Yes

Reviewer #2: Yes

Reviewer #3: None

PLOS authors have the option to publish the peer review history of their article (what does this mean?). If published, this will include your full peer review and any attached files.

**Do you want your identity to be public for this peer review?** For information about this choice, including consent withdrawal, please see our Privacy Policy.

Reviewer #1: **Yes:**Jan Fousek

Reviewer #2: **Yes:**Damien Depannemaecker

Reviewer #3: **Yes:**Leon Stefanovski

**Figure resubmission:**
---

## [Decision Letter · Decision Letter 2]

27 Apr 2026

Dear Dr Ibáñez,

We are pleased to inform you that your manuscript 'A multi-frequency whole-brain neural mass model with homeostatic feedback inhibition' has been provisionally accepted for publication in PLOS Computational Biology.

Best regards,

Sacha Jennifer van Albada

Academic Editor

PLOS Computational Biology

Hugues Berry

Section Editor

PLOS Computational Biology

Reviewer's Responses to Questions

**Comments to the Authors:**

Reviewer #2: All of my comments have been adequately addressed, and I believe the manuscript’s quality and impact have significantly improved compared to the initial version. I have no additional questions or remarks.

**Have the authors made all data and (if applicable) computational code underlying the findings in their manuscript fully available?**

Reviewer #2: None

PLOS authors have the option to publish the peer review history of their article (what does this mean?). If published, this will include your full peer review and any attached files.

Reviewer #2: **Yes:**Damien Depannemaecker

---

## [Editor Report · Acceptance letter]

PCOMPBIOL-D-25-01699R2

A multi-frequency whole-brain neural mass model with homeostatic feedback inhibition

Dear Dr Ibáñez,

I am pleased to inform you that your manuscript has been formally accepted for publication in PLOS Computational Biology. Your manuscript is now with our production department and you will be notified of the publication date in due course.

With kind regards,

Anita Estes
